# Fine-Tuning is Fine, if Calibrated

**Zheda Mai**[1]*, **Arpita Chowdhury**[1]*, **Ping Zhang**[1]*, **Cheng-Hao Tu**[1], **Hong-You Chen**[1],
**Vardaan Pahuja**[1], **Tanya Berger-Wolf**[1], **Song Gao**[2], **Charles Stewart**[3], **Yu Su**[1], **Wei-Lun Chao**[1]
[1]The Ohio State University, [2]University of Wisconsin Madison, [3] Rensselaer Polytechnic Institute.

## Abstract

Fine-tuning is arguably the most straightforward way to tailor a pre-trained model (*e.g.*, a foundation model) to downstream applications, but it also comes with the risk of losing valuable knowledge the model had learned in pre-training. For example, fine-tuning a pre-trained classifier capable of recognizing a large number of classes to master a subset of classes at hand is shown to drastically degrade the model's accuracy in the other classes it had previously learned. As such, it is hard to further use the fine-tuned model when it encounters classes beyond the fine-tuning data. In this paper, we systematically dissect the issue, aiming to answer the fundamental question, *"What has been damaged in the fine-tuned model?"* To our surprise, we find that the fine-tuned model neither forgets the relationship among the other classes nor degrades the features to recognize these classes. Instead, the fine-tuned model often produces more discriminative features for these other classes, even if they were missing during fine-tuning! What really hurts the accuracy is the discrepant logit scales between the fine-tuning classes and the other classes, implying that a simple post-processing calibration would bring back the pre-trained model's capability and at the same time unveil the feature improvement over all classes. We conduct an extensive empirical study to demonstrate the robustness of our findings and provide preliminary explanations underlying them, suggesting new directions for future theoretical analysis. Our code is available at `https://github.com/OSU-MLB/Fine-Tuning-Is-Fine-If-Calibrated`.

## 1 Introduction

Pre-trained models (*e.g.*, foundation models) have become an indispensable component in modern AI development [2]. Building upon neural networks with millions if not billions of parameters and trained with gigantic amounts of data, these models have led to groundbreaking results in various domains [30, 32, 39] and shown several emerging capabilities not observed priorly [21, 27, 2].

Yet, to obtain superior downstream performance, fine-tuning is still often needed. Typically, fine-tuning optimizes the model's performance on the available downstream data. Taking image classification as an example, end-users typically fine-tune the pre-trained classifier to maximize the accuracy of a certain set of classes at hand, no matter whether they know up front that it is the complete set of classes or not. As a result, it is hard to further apply the model to some other classes, *even if the model had learned about those classes in pre-training.* (Please see Figure 1 for an illustration.)

A recent work by Tu *et al.* [49] systematically evidenced such a problem. They fine-tuned a pre-trained classifier with a subset of classes it had learned (*i.e.*, *fine-tuning* classes) and found this led to a drastic accuracy drop in the other classes (*i.e.*, *absent* classes). Tu *et al.* [49] viewed this as an instance of catastrophic forgetting [12, 24, 7] and suggested two ways to address it: (1) identifying the model updating direction that can improve both the fine-tuning and absent classes; (2) preserving class relationships. They presented a strong baseline, combining a new stochastic gradient descent (SGD)

---

*Equal contributions.

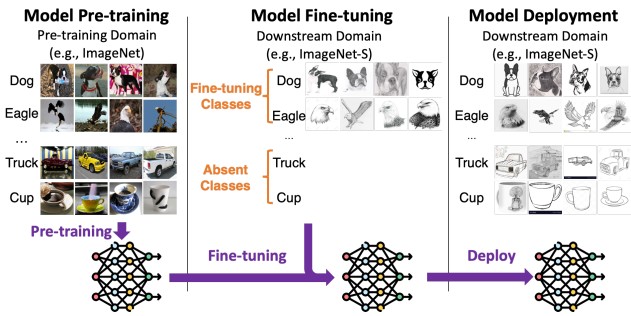
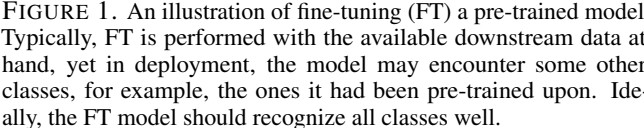

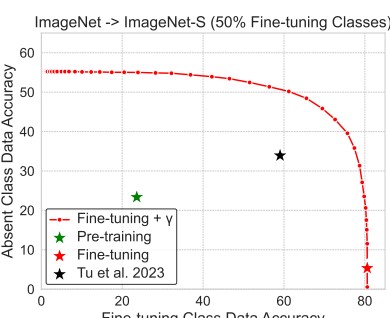

FIGURE 1. An illustration of fine-tuning (FT) a pre-trained model. Typically, FT is performed with the available downstream data at hand, yet in deployment, the model may encounter some other classes, for example, the ones it had been pre-trained upon. Ideally, the FT model should recognize all classes well.

FIGURE 2. **Fine-tuning** ($\star$) + **post-processing calibration** with a bias factor $\gamma$ ($-\cdot-$) can outperform the SOTA solution ($\star$) [49].

strategy, feature rank regularization [6, 46], distillation [16], and frozen classifiers [29], achieving decent results on preserving absent class accuracy while improving fine-tuning class accuracy.

We build upon [49] to further analyze the problem of *fine-tuning a pre-trained model*. Specifically, we want to understand *which part of the pre-trained classifier or what knowledge it had learned* is damaged after fine-tuning with a subset of classes.

We first analyze the feature extractor of the fine-tuned model. *If the fine-tuned feature extractor forgets the absent classes, the extracted features would exhibit poor discrimination among these classes or confuse them with the fine-tuning classes.* We apply the Nearest Class Mean (NCM) classifier [5, 44, 55] to assess the feature quality after fine-tuning, using the hold-out data for calculating class means. Surprisingly, the fine-tuned feature extractor does not degrade but often improves for the absent classes! The resulting features could better separate absent-class data and raise the NCM classification accuracy over all the classes. This finding suggests that fine-tuning with merely a subset of classes can already *holistically* improve the feature extractor for the downstream domain.

To search for the root cause of accuracy drops, we analyze the fine-tuned classifier as a whole but *decompose its prediction rule into two parts*: a binary classifier separating fine-tuning and absent classes, followed by the multi-class classifier dedicated to each set of classes. We find that the fine-tuned model can distinguish among the absent classes very well, implying that the absent-class relationship in the fully connected layer is preserved. *The only component that is damaged but ends up failing the whole model is the binary classifier.* Concretely, the biased logits towards fine-tuning classes make most absent-class examples misclassified as fine-tuning classes.

*The above findings are encouraging and have profound implications!* They imply that a simple *post-processing calibration* can potentially address the fine-tuned model's inferior accuracy on the absent classes, bringing back the pre-trained model's capability while unveiling the improved feature quality over all classes. Indeed, by adding a calibration bias factor to all the absent classes' logits [4, 41], the fine-tuned model can successfully reclaim the absent class accuracy and obtain decent overall improvement in the downstream domain (Figure 2). The resulting performance even beats the strong baseline [49] in many of the benchmarks, including ImageNet and its variants [8, 15, 54], Office-Home [53], and VTAB [64], *without complicated training and hyperparameter setting*.

The "unexpected benign behaviors" of fine-tuning with a subset of classes raise several interesting follow-up questions. For example, are there any specific setups in our experiment contributing to the benign behaviors? If not, what are the explanations underlying these benign behaviors?

We consider different splits of fine-tuning and absent classes in terms of their distributions and numbers. We find that the benign behaviors are robust to the number of fine-tuning classes. Even if the fine-tuning and absent classes are semantically distinct (*e.g.*, animals as fine-tuning and vehicles as absent classes), the absent class accuracy after fine-tuning stays quite close to that of the pre-trained model without suffering catastrophic forgetting. We further investigate different optimizers for fine-tuning. When the SGD optimizer is used, the benign behaviors are robust to hyperparameters such as learning rates and weight decay. When more advanced, adaptive optimizers like Adam [23] are applied, we observe noticeable degradation with improperly selected hyperparameters. Still, with smaller enough learning rates and weight decay, the benign behaviors show up and hold.

We conduct further analyses to understand the benign behaviors. Specifically, we investigate how fine-tuning with a subset of classes impacts the other (*i.e.*, absent) class features. Our derivation shows that after each SGD step, the feature update of an absent-class example is governed by the gradients w.r.t. its most similar fine-tuning class examples. Suppose the gradients w.r.t. fine-tuning class data could effectively capture the downstream-specific information, our derivation offers a preliminary explanation of why fine-tuning with a subset of classes could improve the other, absent class features in the downstream domain. Please refer to section 6 for details and other analyses regarding the class relationship, frozen classifiers, etc.

**Remark.** We systematically dissect the damage caused by fine-tuning a pre-trained classifier with a subset of classes at hand. Our insights suggest that a simple post-processing calibration is sufficient to mitigate the damage, reclaiming the pre-training model's capability while holistically improving the accuracy in the downstream domain. Our study also opens up several interesting questions worth further exploration. For example, it is intriguing that end-to-end fine-tuning of the whole model without additional regularization and intermediate supervision only degrades a part of the model.

We note that NCM and post-processing calibration are well-known machine learning techniques and we certainly do not claim them as our novelties. Instead, we use them because we find their respective properties and application scenarios suitable for our problem. We use NCM to assess feature qualities because its performance solely depends on features, not the last fully connected layer. We identify post-processing calibration as a promising solution because we find that *the only major damage in the fine-tuned model is the biased logits towards fine-tuning classes*. In other words, if the fine-tuned model also suffers from feature degradation or class relationship forgetting (to our surprise, it does not), calibration alone is unlikely sufficient to address the problem.

## 2   Related Work

**Fine-tuning.** The basic methods are linear probing and full fine-tuning [26]. Parameter-efficient fine-tuning (PEFT) [60, 35, 50] has attracted increasing attention lately (mainly for Transformers [52]), aiming to update only a fraction of pre-trained parameters on top of linear probing. *We focus on full fine-tuning* because it is model-agnostic and arguably still the most widely used method. That said, we expect the insights and implications from our study not to be limited to full fine-tuning and potentially transferable to other fine-tuning methods as well.

**Risk in fine-tuning.** When the downstream data is scarce, fine-tuning is prone to over-fitting and needs certain forms of regularization [28]. Even with sufficient data to represent the downstream task, fine-tuning may risk losing valuable knowledge the pre-trained model had learned. For example, [57] showed that fine-tuning with data from a specific domain (*e.g.*, ImageNet-1K real images) degrades the pre-trained model's generalizability to other domains (*e.g.*, ImageNet-Sketch/Rendition). On an orthogonal dimension, [49] showed that fine-tuning a pre-trained classifier with a subset of classes it had learned led to a huge accuracy drop in the other classes. Similar phenomenons have been observed in [66, 40]. Along with these findings come several proposed solutions. [57] showed that a weight interpolation between the pre-trained and fine-tuned models reclaims the pre-trained model's generalizability. [49] investigated many approaches, including weight interpolation, but found they are insufficient to preserve the accuracy in the other classes. They presented a novel SGD strategy that helps identify the gradient direction benefiting both fine-tuning and other classes. [66] developed dedicated solutions to CLIP-based vision-language models to preserve accuracy for absent classes.

*We 1) consider the fine-tuning scenario studied in [49] and 2) use the standard neural network classifier architecture with a fully connected layer on top.* Our focus is not to propose a brand-new solution and compete with existing ones, but to understand the underlying cause of the accuracy drop.

**Continual learning and catastrophic forgetting.** *Fine-tuning a pre-trained model with a subset of classes it had learned* is related to continual learning [7] but has several differences. Class-incremental learning [33, 36] aims to expand the model's label space. In contrast, we suppose the pre-trained model had learned a wide range of classes; fine-tuning is meant to tailor it to a downstream domain (*e.g.*, a different image style), not to learn extra categories. Compared to domain-incremental learning [33, 31], we do not ask the fine-tuned model to retain its performance in the pre-training domain. That said, the accuracy drop observed in our study and continual learning can all be considered certain forms of catastrophic forgetting [37, 12]. Indeed, a recent survey by Wang *et al*. [56] argued that forgetting is a common issue in various research fields, including foundation models, meta-learning,

test-time adaptation, and generative models, among others. We thus expect our findings to serve as a valuable reference for addressing the forgetting issue resulting from fine-tuning.

**Post-processing calibration.** Training with class-imbalanced data is known to produce biased logits towards major classes [61]. Such an issue not only appears in long-tailed recognition [20, 18] but also few-shot learning [62, 45] and continual learning with replay buffers [48, 47, 34]. Post-processing calibration [3, 38, 58, 59] is a widely applicable method to address this issue, which adjusts model confidence during inference. Popular methods include normalizing classifier norms [20, 17] and adjusting logits according to class sizes [61, 38, 22, 65]. Unlike the machine learning problems above, whether or not post-processing calibration can address the accuracy drop in our problem is not immediately clear. Without access to the absent class data, fine-tuning may have ruined the features or linear classifiers corresponding to these classes. If so, simply performing post-processing calibration cannot reclaim the accuracy of the absent classes. Our main contribution is therefore not merely the solution, but the systematic study that identifies post-processing calibration as an effective solution.

**Other paradigms.** There are several other machine learning paradigms related to the fine-tuning setting we study. We refer the readers to [49] for an in-depth discussion and comparison.

## 3 Background

**Problem definition.** We study the problem of fine-tuning a pre-trained classifier capable of recognizing $C$ classes, using data from a subset of $C^\star$ classes at hand. The goal is to tailor the classifier to the downstream domain (*e.g.*, a different image style).

More formally, let us denote by $D_{\text{tr}} = \{(\boldsymbol{x}_i, y_i \in \mathcal{S})\}_{i=1}^N$ the data set for fine-tuning, where $\mathcal{S}$ is a **strict subset** of the pre-trained model's label space $\mathcal{Y}$, *i.e.*, $|\mathcal{S}| = C^\star < C = |\mathcal{Y}|$. Let us denote a neural network classifier by $\hat{y} = \arg\max_{c \in \mathcal{Y}} \boldsymbol{w}_c^\top f_{\boldsymbol{\theta}}(\boldsymbol{x})$, where $\boldsymbol{x}$ is an input sample, $f_{\boldsymbol{\theta}}$ is the feature extractor parameterized by $\boldsymbol{\theta}$, and $\boldsymbol{W} = \{\boldsymbol{w}_c\}_{c=1}^C$ is the set of linear classifiers, a.k.a., the fully-connected (FC) layer. We call $\{\boldsymbol{\theta}, \boldsymbol{W}\}$ the model parameters. The value $\boldsymbol{w}_c^\top f_{\boldsymbol{\theta}}(\boldsymbol{x})$ is often referred to as the decision value or logit of class $c$.

Without loss of generality, we define $\mathcal{Y} = \{1, \cdots, C\}$ and $\mathcal{S} = \{1, \cdots, C^\star\}$. That is, they share the first $C^\star$ classes. We call $\mathcal{S}$ the fine-tuning classes and use $\mathcal{U} = \{(C^\star + 1), \cdots, C\}$ to denote the absent classes during fine-tuning, where $\mathcal{S} \cap \mathcal{U} = \emptyset$ and $\mathcal{S} \cup \mathcal{U} = \mathcal{Y}$.

**Fine-tuning and its issue.** Full fine-tuning updates the pre-trained model $\{\boldsymbol{\theta}_{\mathcal{O}}, \boldsymbol{W}_{\mathcal{O}}\}$ by

$$\{\boldsymbol{\theta}_{\mathcal{T}}, \boldsymbol{W}_{\mathcal{T}}\} = \arg\min_{\{\boldsymbol{\theta}, \boldsymbol{W}\}} \sum_{i=1}^N \ell(\boldsymbol{x}_i, y_i; \boldsymbol{\theta}, \boldsymbol{W}), \qquad \text{with } \{\boldsymbol{\theta}, \boldsymbol{W}\} \text{ initialized by } \{\boldsymbol{\theta}_{\mathcal{O}}, \boldsymbol{W}_{\mathcal{O}}\},$$

where $\ell$ denotes the cross-entropy loss; $\{\boldsymbol{\theta}_{\mathcal{T}}, \boldsymbol{W}_{\mathcal{T}}\}$ denotes the fine-tuned model.

Since $D_{\text{tr}}$ only covers a subset of classes $\mathcal{S}$, the fine-tuned model $\{\boldsymbol{\theta}_{\mathcal{T}}, \boldsymbol{W}_{\mathcal{T}}\}$ was observed to degrade drastically in classifying data from the absent classes $\mathcal{U}$ [49]. Figure 2 shows one example: the absent class accuracy in the y-axis drops from $\sim 23\%$ (⋆) to only $\sim 3\%$ (marked ⋆) after fine-tuning, even though the fine-tuning class accuracy in the x-axis increases hugely by roughly $60\%$.

**Terminology.** Tu *et al*. named the above setting Holistic Transfer (HT) [49]. The core challenge is how to maintain and even improve the fine-tuned model's ability to recognize the $C - C^\star$ absent classes in the downstream domain. For ease of reference, we use the following terms interchangeably.

- fine-tuning classes & seen classes; absent classes & unseen classes;
- downstream domain & target domain; pre-training domain & source domain;
- fine-tuning & naive fine-tuning.

We emphasize that the "unseen" classes are indeed seen in pre-training but absent in fine-tuning.

## 4 A Systematic Study of Fine-Tuning (FT)

Seeing the ability to classify the absent classes "disappears" after fine-tuning, we are curious about

- how each component of the fine-tuned model $\{\boldsymbol{\theta}_{\mathcal{T}}, \boldsymbol{W}_{\mathcal{T}}\}$ contributes to it;

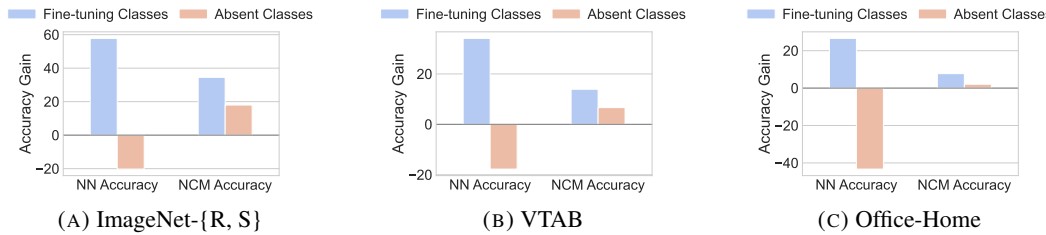

| (A) ImageNet-{R, S} | (B) VTAB | (C) Office-Home |

FIGURE 3. **Accuracy gain after fine-tuning.** We consider the neural network (NN) classifier with the FC layer (section 3) and the NCM classifier using only features (Equation 2). We show the average accuracy gain on fine-tuning classes ($\mathrm{Acc}_{\mathcal{S}/\mathcal{Y}}$) and absent classes ($\mathrm{Acc}_{\mathcal{U}/\mathcal{Y}}$). While the NN classifier suffers drops in $\mathrm{Acc}_{\mathcal{U}/\mathcal{Y}}$, the NCM classifier enjoys a consistent gain, suggesting the holistic improvement of features after fine-tuning.

- whether the ability is "forgotten" forever or "buried" temporarily by some other damaging factors emerging during fine-tuning.

To this end, we conduct a systematic analysis, aiming to dissect the degradation caused by fine-tuning.

### 4.1 Experiment setup: datasets, models, and evaluation metrics

We focus on two of the largest datasets used in [49]. We also consider the ImageNet Distribution Shift benchmark widely used in out-of-distribution (OOD) generalization [57].

1. **Office-Home** [53]: a domain adaptation dataset with 65 classes and 4 domains. For each source-target pair, we pre-train a ResNet-50 [14] using the source data and fine-tune it on 30 randomly selected classes in the target domain. We evaluate the model on all 65 classes in the target domain.
2. **VTAB** [64]: a set of 19 visual recognition datasets. We follow [49] to use the 8 datasets provided with text labels and use CLIP (ViT-B/32) [42] as the pre-trained model. We extract the class name embedding to construct the FC layer and discard the CLIP text encoder afterward. We fine-tune the model on the randomly selected 50% of classes in each dataset and test it on all the classes.
3. **ImageNet-R [15]** and **ImageNet-S [54]**: datasets for OOD detection and generalization [57]. ImageNet-R [15] consists of 200 ImageNet classes [8, 43] with renditions (paintings, cartoons, etc.). ImageNet-S [54] consists of 1K ImageNet classes with sketching. We use ResNet-50 (ViT-B/32 [9] results in Appendix D) pre-trained on ImageNet-1K. We fine-tune them on randomly sampled 50% of classes (100/500 for ImageNet-R/S, respectively) and test them on all the classes.

By default, we use the cross-entropy loss and optimize the model using the SGD momentum optimizer.

**Evaluation metric.** Let $D_{\mathrm{tr}} = \{(\boldsymbol{x}_i, y_i \in \mathcal{Y})\}_{j=1}^{M}$ denote the downstream test set covering all the classes, we define the accuracy of classifying data belonging to classes $\mathcal{A}$ into the label space $\mathcal{B}$ by

$$\mathrm{Acc}_{\mathcal{A}/\mathcal{B}} = \frac{\sum_i \mathbb{1}[y_i \in \mathcal{A}] \times \mathbb{1}[y_i = \arg\max_{c \in \mathcal{B}} \ \boldsymbol{w}_c^\top f_{\boldsymbol{\theta}}(\boldsymbol{x}_i)]}{\sum_i \mathbb{1}[y_i \in \mathcal{A}]}. \tag{1}$$

For instance, $\mathrm{Acc}_{\mathcal{S}/\mathcal{Y}}$ is the accuracy of classifying fine-tuning class data into all $C$ classes; $\mathrm{Acc}_{\mathcal{U}/\mathcal{Y}}$ is the accuracy of classifying absent class data into all $C$ classes. We note that these are the two accuracies reported in [49] and depicted in Figure 2. In this paper, we further consider $\mathrm{Acc}_{\mathcal{S}/\mathcal{S}}$ and $\mathrm{Acc}_{\mathcal{U}/\mathcal{U}}$, corresponding to classifying each set of data into their respective label space.

### 4.2 Is the fine-tuned feature extractor damaged?

We first investigate the feature extractor $f_{\boldsymbol{\theta}}$, *i.e.*, whether the fine-tuned extractor $f_{\boldsymbol{\theta}_{\mathcal{T}}}$ forgets the discriminative ability to differentiate absent-class samples. We apply the NCM classifier [5, 44, 55], whose accuracy is solely governed by the feature quality, to examine the feature extractor. Given a test example $\boldsymbol{x}$, the NCM classification rule is

$$\hat{y} = \arg\min_{c \in \mathcal{B}} \left\| \frac{f_{\boldsymbol{\theta}_{\mathcal{T}}}(\boldsymbol{x})}{\|f_{\boldsymbol{\theta}_{\mathcal{T}}}(\boldsymbol{x})\|_2} - \boldsymbol{\mu}_c \right\|_2, \tag{2}$$

which outputs the class $\hat{y} \in \mathcal{B}$ whose feature mean $\boldsymbol{\mu}_{\hat{y}}$ is the closest to $f_{\boldsymbol{\theta}}(\boldsymbol{x})$. We hold out a subset of the downstream data to calculate the class mean, for both fine-tuning and absent classes.

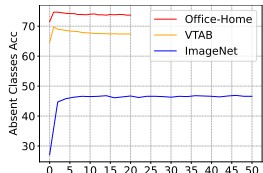

FIGURE 4. $\text{Acc}_{\mathcal{U}/\mathcal{U}}$ along with the fine-tuning epochs.

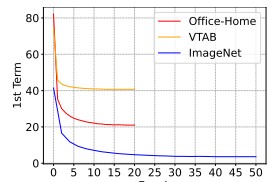

FIGURE 5. Average predicted probability that absent class examples belong to absent classes.

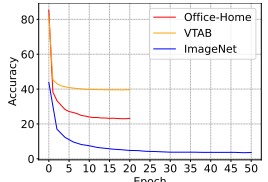

FIGURE 6. % of absent class examples classified as absent classes.

We compute the NCM accuracy (with $\mathcal{B} = \mathcal{Y}$) on the fine-tuning class and absent class data using the fine-tuned (FT) feature extractor $f_{\boldsymbol{\theta}_{\mathcal{T}}}$, *i.e.*, $\text{Acc}_{\mathcal{S}/\mathcal{Y}}(f_{\boldsymbol{\theta}_{\mathcal{T}}})$ and $\text{Acc}_{\mathcal{U}/\mathcal{Y}}(f_{\boldsymbol{\theta}_{\mathcal{T}}})$. We also apply the pre-trained feature extractor $f_{\boldsymbol{\theta}_{\mathcal{O}}}$ and obtain $\text{Acc}_{\mathcal{S}/\mathcal{Y}}(f_{\boldsymbol{\theta}_{\mathcal{O}}})$ and $\text{Acc}_{\mathcal{U}/\mathcal{Y}}(f_{\boldsymbol{\theta}_{\mathcal{O}}})$. We report the NCM accuracy gap between the two feature extractors in Figure 3. As one may expect, the FT extractor $f_{\boldsymbol{\theta}_{\mathcal{T}}}$ improves the fine-tuning class accuracy. However, surprisingly, it also improves the absent class accuracy *without catastrophic forgetting*. This result sharply contrasts the accuracy obtained by the full neural network (NN) classifier (cf. section 3). As shown in Figure 3, the absent class accuracy by the NN classifier drops by $15 \sim 40\%$ after fine-tuning. In short, we find that fine-tuning with a subset of classes can adapt the feature extractor holistically to the downstream domain to improve both the fine-tuning and absent classes, capturing the gradient direction of the domain shift [49].

**Remark.** We use NCM *only* to assess feature qualities, not as the final classifier. After all, we do not have absent class data to compute the class means. This contrasts the use of NCM in continual or few-shot learning, where the prior- or few-shot class data are accessible (*e.g.*, through replay buffers).

We emphasize that the feature improvement here is not as trivial as observed in conventional transfer or few-shot learning. **In these learning problems**, the pre-trained model typically had not learned about the absent classes; thus, there is no risk of forgetting them. Besides, the fine-tuning and absent classes are often treated as two separate tasks in evaluation; thus, it is unclear whether the fine-tuned feature extractor improves the overall classification accuracy. **In contrast, in our fine-tuning setting**, the pre-trained model had already learned about the absent classes. Fine-tuning thus risks forgetting them. However, as shown in Figure 3, fine-tuning improves absent class accuracy, in the context where the fine-tuning and absent classes are evaluated together in a single task.

### 4.3 What is damaged in the fine-tuned neural network classifier?

The findings in subsection 4.2 eliminate the feature extractor $\boldsymbol{\theta}_{\mathcal{T}}$ from the candidate root causes of the degraded FT model $\{\boldsymbol{\theta}_{\mathcal{T}}, \boldsymbol{W}_{\mathcal{T}}\}$. The drastic drop of absent class accuracy thus must come from the FC layer $\boldsymbol{W}_{\mathcal{T}}$ or its alignment with the feature extractor. To analyze what goes wrong, we decompose the softmax probability induced by the neural network classifier as follows,

$$p(c|\boldsymbol{x}) = \frac{\exp\left(\boldsymbol{w}_c^\top f_{\boldsymbol{\theta}}(\boldsymbol{x})\right)}{\sum_{c' \in \mathcal{Y}} \exp\left(\boldsymbol{w}_{c'}^\top f_{\boldsymbol{\theta}}(\boldsymbol{x})\right)} = \frac{z_c(\boldsymbol{x})}{\sum_{c' \in (\mathcal{S} \cup \mathcal{U})} z_{c'}(\boldsymbol{x})} \tag{3}$$

$$= \frac{\sum_{c' \in \mathcal{U}} z_{c'}(\boldsymbol{x})}{\sum_{c' \in \mathcal{S}} z_{c'}(\boldsymbol{x}) + \sum_{c' \in \mathcal{U}} z_{c'}(\boldsymbol{x})} \times \frac{z_c(\boldsymbol{x})}{\sum_{c' \in \mathcal{U}} z_{c'}(\boldsymbol{x})}, \tag{4}$$

where $z_c(\boldsymbol{x}) = \exp\left(\boldsymbol{w}_c^\top f_{\theta}(\boldsymbol{x})\right)$. Let $c$ be an absent class, the $1^{\text{st}}$ term stands for the predicted probability that $\boldsymbol{x}$ belongs to the absent classes $\mathcal{U}$, not the fine-tuning classes $\mathcal{S}$; the $2^{\text{nd}}$ term stands for the probability that within the absent classes $\mathcal{U}$, $\boldsymbol{x}$ belongs to class $c$. Correctly classifying an absent class example $(\boldsymbol{x}, y \in \mathcal{U})$, *i.e.*, $y = \arg\max_{c \in \mathcal{Y}} p(c|\boldsymbol{x})$, thus requires 1) obtaining a high probability in the $1^{\text{st}}$ term and 2) correctly classifying the example among absent classes.

Building upon this insight, we analyze the accuracy of taking $\arg\max$ of the $2^{\text{nd}}$ term to classify absent class examples. We note that this is exactly the $\text{Acc}_{\mathcal{U}/\mathcal{U}}$ defined in subsection 4.1. As shown in Figure 4, $\text{Acc}_{\mathcal{U}/\mathcal{U}}$ does not degrade but improves in the first few epochs of fine-tuning and stays stable afterward. This result implies two key messages. First, the FT model does not forget its ability to distinguish among absent classes. Second, the drastic accuracy drop of $\text{Acc}_{\mathcal{U}/\mathcal{Y}}$ results from the $1^{\text{st}}$ term in Equation 4 — the binary classifier separating fine-tuning and absent classes.

As shown in Figure 5, the predicted probability that *absent class examples belong to absent classes* (*i.e.*, the average of the $1^{\text{st}}$ term over absent class examples) reduces notably along with the fine-tuning

| Metrics (%) | ImageNet-{R, S} | | | VTAB | | | Office-Home | | |
|---|---|---|---|---|---|---|---|---|---|
| | $\text{Acc}_{\mathcal{Y}/\mathcal{Y}}$ | $\text{Acc}_{\mathcal{S}/\mathcal{Y}}$ | $\text{Acc}_{\mathcal{U}/\mathcal{Y}}$ | $\text{Acc}_{\mathcal{Y}/\mathcal{Y}}$ | $\text{Acc}_{\mathcal{S}/\mathcal{Y}}$ | $\text{Acc}_{\mathcal{U}/\mathcal{Y}}$ | $\text{Acc}_{\mathcal{Y}/\mathcal{Y}}$ | $\text{Acc}_{\mathcal{S}/\mathcal{Y}}$ | $\text{Acc}_{\mathcal{U}/\mathcal{Y}}$ |
| Pre-trained | 23.6 | 23.5 | 23.8 | 58.3 | 59.3 | 57.4 | 63.8 | 61.8 | 65.5 |
| Fine-tuning | 43.3 | 81.3 | 3.5 | 62.8 | 86.4 | 39.1 | 53.5 | 88.3 | 22.4 |
| Tu et al. [49] | 47.5 | 62.1 | 23.1 | 63.8 | 83.9 | 43.5 | 72.0 | 78.2 | 66.6 |
| Fine-tuning + $\gamma_{\text{ALG}}$ | 55.9 | 80.3 | 30.5 | 66.8 | 85.3 | 48.2 | 65.0 | 87.7 | 44.9 |
| Fine-tuning + $\gamma_{\text{PCV}}$ | 57.1 | 60.1 | 54.0 | 57.4 | 47.1 | 67.8 | 72.2 | 82.3 | 63.1 |
| Fine-tuning + $\gamma\star$ | 60.8 | 73.6 | 47.6 | 69.3 | 75.6 | 62.8 | 72.7 | 79.1 | 66.9 |
| Oracle | 71.1 | 72.4 | 69.8 | 80.6 | 79.8 | 81.3 | 82.1 | 81.2 | 82.9 |

TABLE 1. Post-processing calibration can effectively bring back the pre-trained model's capability in recognizing absent classes. Oracle is based on a classifier fine-tuned with both fine-tuning and absent class data.

epochs, suggesting the tendency to misclassify an absent class sample as fine-tuning classes. Indeed, as shown in Figure 6, for most of the absent class samples, the FT model ends up producing a lower value of $\max_{c\in\mathcal{U}} \boldsymbol{w}_c^\top f_\theta(\boldsymbol{x})$ than $\max_{c\in\mathcal{S}} \boldsymbol{w}_c^\top f_\theta(\boldsymbol{x})$, misclassifying them into fine-tuning classes. In short, *the main factor that damages the FT model's ability to correctly classify absent class examples is the biased logit values towards fine-tuning classes.*

## 5 Post-Processing Calibration for the Rescue

The systematic study in section 4 highlights several key characteristics of the FT model $\{\boldsymbol{\theta}_\mathcal{T}, \boldsymbol{W}_\mathcal{T}\}$. First, the model retains and even improves the accuracy of classifying absent class samples when the label space is limited to absent classes (*i.e.*, $\text{Acc}_{\mathcal{U}/\mathcal{U}}$). Second, the model tends to assign much higher decision values $\boldsymbol{w}_c^\top f_\theta(\boldsymbol{x})$, a.k.a. logits, to the fine-tuning classes, ending up misclassifying most absent class samples into fine-tuning classes and thus hurting $\text{Acc}_{\mathcal{U}/\mathcal{Y}}$.

*These characteristics suggest that a simple, post-processing calibration of the FT model's logits could potentially bring back the pre-trained model's ability to classify absent classes correctly.* To this end, we apply the calibration approach proposed in a different context of zero-shot learning [4], which adds a factor $\gamma$ uniformly to the logits of all absent classes, leading to a new classification rule

$$\hat{y} = \arg\max_{c\in\mathcal{Y}} \quad \boldsymbol{w}_c^\top f_\theta(\boldsymbol{x}) + \gamma\mathbb{1}[c \in \mathcal{U}]. \tag{5}$$

This calibration factor lifts the logits of absent classes to a level comparable with those of fine-tuning classes. Suppose an absent class example is correctly classified among absent classes but misclassified into fine-tuning classes, adding a sufficiently large $\gamma$ could reclaim the correct prediction.

We design two approaches to properly set $\gamma$ **without** accessing the absent class data. **Average logit gap (ALG)** estimates $\gamma$ by the average logit gap between non-ground-truth fine-tuning classes and absent classes in the fine-tuning data $D_{\text{tr}}$. **Pseudo cross-validation (PCV)** partitions $D_{\text{tr}}$ into pseudo-fine-tuning and pseudo-absent classes and finds $\gamma$ that can balance the pseudo-fine-tuning and pseudo-absent class accuracy. We also investigate a **"cheating"** $\gamma\star$ based on the test data to estimate the upper-bound results. More details are in Appendix B.

**Performance.** Table 1 shows the results on ImageNet-{R, S}, VTAB, and Office-Home, averaged over datasets within each benchmark. Despite its simplicity, calibration effectively boosts the FT model's accuracy on the absent classes, achieving comparable accuracy to the SOTA method proposed in [49]. On ImageNet-{R, S}, both calibration approaches outperform the SOTA by a notable margin.

**Compatibility of calibration.** Post-processing calibration is potentially applicable to any model. In Figure 7, we extend the calibration approach to the pre-trained model and the SOTA model [49] with varying $\gamma$. Calibration can effectively balance $\text{Acc}_{\mathcal{S}/\mathcal{Y}}$ and $\text{Acc}_{\mathcal{U}/\mathcal{Y}}$ and trade one for the other. Interestingly, we find the curve of the FT model, in most cases, can well cover those of the other models. To characterize this, we follow [4]

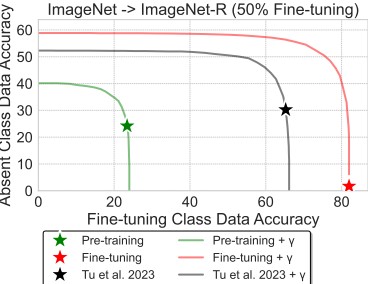

FIGURE 7. **Fine-tuning vs. Absent Class Accuracy Curve** ($\text{Acc}_{\mathcal{S}/\mathcal{Y}}$ at the x-axis; $\text{Acc}_{\mathcal{U}/\mathcal{Y}}$ at the y-axis) by varying the factor $\gamma$ for ImageNet-R.

| AUSUC | IN-{R, S} | VTAB | O-H |
|---|---|---|---|
| Pre-trained | 0.083 | 0.444 | 0.499 |
| Tu et al. [49] | 0.317 | 0.553 | 0.618 |
| Fine-tuning | **0.439** | **0.586** | **0.632** |

TABLE 2. Results in AUSUC.

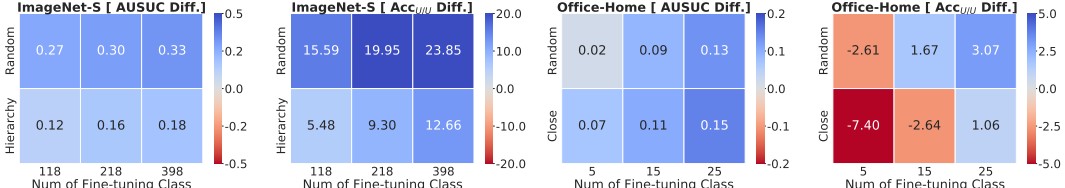

FIGURE 8. The performance gain on AUSUC and NCM $\text{Acc}_{\mathcal{U}/\mathcal{Y}}$ (from the pre-trained model to the FT model), under different data splits and fine-tuning class sizes. The hierarchical split of ImageNet-S contains dog (118 classes), mammal (218 classes), and animal (398 classes) as the fine-tuning classes.

to calculate the Area Under the Seen-Unseen ((Fine-tuning / Absent)) Curve (AUSUC) to summarize the overall performance of each model across the entire spectrum of $\gamma$. Table 2 reports the AUSUC. The FT model notably outperforms the other methods, showing its robustness in learning from a subset of classes.

**Remark.** Many post-processing calibration methods have been proposed (cf. section 2). Our study is not meant to compare them, but to show that calibration can effectively address the issue caused by fine-tuning. We also note that Tu *et al*. [49] has investigated many solutions, including weight interpolation [57], but found them less effective them their proposed SOTA method.

## 6 Ablation Study and Additional Analysis

**Data split and fine-tuning class size.** In the default setup of [49], fine-tuning classes and absent classes are uniformly randomly sampled and have each portion close to $50\%$. In practice, however, end-users may have a smaller number of classes at hand or collect data from semantically or conceptually similar classes whose appearances are positively correlated. To explore such practical situations, we investigate how 1) a biased sampling such that the fine-tuning classes are conceptually or distributionally similar to each other than the absent classes and 2) a smaller size of fine-tuning classes would impact the performance of fine-tuning. Specifically, for Office-Home, classes that are similar in the pre-trained model's feature space are selected as fine-tuning classes, leaving the rest as absent ones. We also vary the number of fine-tuning classes. For ImageNet-S, we leverage the WordNet [11] hierarchy to select coherent groups, such as dog (118 classes) and mammal (218 classes) as the fine-tuning classes. Please see Appendix A for details.

Figure 8 illustrates the performance gain on AUSUC and NCM $\text{Acc}_{\mathcal{U}/\mathcal{Y}}$ (from the pre-trained model to the FT model). Notably, the hierarchical split poses a greater challenge for fine-tuning in transferring domain knowledge from fine-tuning classes to absent classes, as evidenced by its relatively minor AUSUC improvements compared to the random split. This difficulty is attributed to the inherent challenge of transferring features learned from dogs or animals to distinctly different classes like TVs and trucks. Furthermore, smaller fine-tuning class sizes present additional difficulties, leading to less pronounced improvements, especially in Office-Home. In summary, our analysis suggests that the benign behaviors of fine-tuning are robust in more practical and difficult splits but its performance requires further improvement when the fine-tuning class size is exceptionally small. Detailed experimental setup and additional results are available in Appendix D.

**Optimizer.** Prior work [49] predominantly used the SGD optimizer. To investigate optimizers' influence in fine-tuning, we scrutinize six popular optimizers including SGD with Momentum, Adam [23], AdaBelief [67], Adadelta [63], AdaGrad [10] and RMSprop. Our study includes 1) a variation across learning rates (LR), adjusted as multipliers of the default ones, *i.e*., $[10, 1, 0.1, 0.01] \times$ default LR, and 2) three distinct weight decays: $[0, 5e - 4, 5e - 3]$. The AUSUC, as illustrated in Figure 9, reveal the robustness of the benign behaviors to different hyperparameter settings of the SGD optimizer. Conversely, more advanced, adaptive optimizers show higher sensitivity when hyperparameters are not properly chosen. Nevertheless, under small enough learning rates (and weight decay), they perform similarly to SGD and notably improve the pre-trained model (whose AUSUC is around $0.5$ in Office-Home).

**Why is absent class relationship preserved?** To elucidate the preservation of the absent class relationship, we analyze how classifier weights $W$ change within the fine-tuning and absent classes. We commence by calculating the L2 normalized weight changes between fine-tuning and the pre-

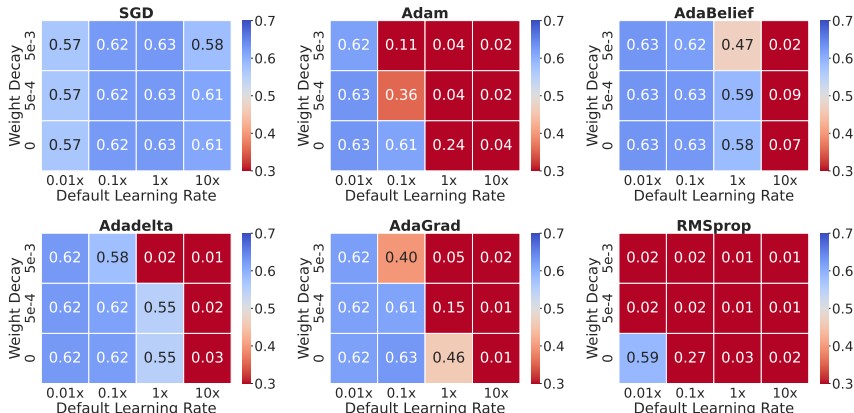

FIGURE 9. Different optimizers and hyperparameters. We use the Office-Home and report the AUSUC. The AUSUC of the pre-trained model is $\sim 0.5$. Fine-tuning with SGD shows remarkable robustness to different learning rates and weight decay. Advanced optimizers necessitate more careful hyperparameter selection. Nevertheless, they perform similarly to SGD under small learning rates and weight decay.

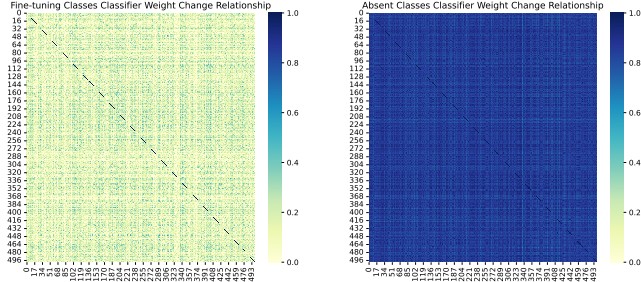

FIGURE 10. Classifier update direction similarity within fine-tuning (left) and absent (right) classes for ImageNet-S. The update directions are highly similar within absent classes, thus preserving the inter-class relationships among absent classes.

trained model $\Delta \boldsymbol{W} = \frac{\boldsymbol{W}_{\mathcal{T}} - \boldsymbol{W}_{\mathcal{O}}}{\|\boldsymbol{W}_{\mathcal{T}} - \boldsymbol{W}_{\mathcal{O}}\|_2}$. Visualizing the cosine similarity of the change within fine-tuning and absent classes, *i.e.* $\Delta \boldsymbol{W}^{\mathcal{Y}} (\Delta \boldsymbol{W}^{\mathcal{Y}})^{\top}$ and $\Delta \boldsymbol{W}^{\mathcal{U}} (\Delta \boldsymbol{W}^{\mathcal{U}})^{\top}$, allows us to assess the directional similarities in classifier updates. As depicted in Figure 10, there exists a pronounced dissimilarity in the update directions among fine-tuning classes since the gradients aim to separate fine-tuning classes during fine-tuning. Conversely, a notable similarity in the update directions is observed among absent classes. The absence of positive signals from absent class data results in nearly uniform updates of absent classifiers, thereby preserving the class relationships among absent classes throughout fine-tuning. More results for other datasets can be found in Appendix D.

**Absent Features and linear classifier Alignment in fine-tuning.** section 4 has demonstrated that the alignment between absent features and the linear classifier (*i.e.*, the FC layer) remains intact during fine-tuning, as evidenced by the stable or improved $\text{Acc}_{\mathcal{U}/\mathcal{U}}$, despite the absence of absent class data. This indicates that fine-tuning retains, and even enhances, its capacity to differentiate among absent classes. To delve deeper into this phenomenon, we scrutinize the behavior of ground-truth class (GT) vs. the largest non-GT absent logits for absent test samples. As depicted in Figure 12, although both sets of logits exhibit a decline throughout the training process— attributable to the lack of absent data during fine-tuning—the relative difference between them persists, underlining a consistent and stable alignment between the features and the linear classifier. Future work may delve into the theoretical understanding of this observed alignment between features and classifiers for absent classes.

**Why do absent class features improve after fine-tuning?** During fine-tuning, the pre-trained model is updated solely by gradients derived from the fine-tuning class data. Surprisingly, the fine-tuned feature extractor does not forget but improves its ability to differentiate the absent class data. Here, we explain it by considering a two-layer linear neural network $\hat{y} = \arg\max_c \boldsymbol{w}_c^{\top} (\boldsymbol{U}\boldsymbol{x}) = \arg\max_c \boldsymbol{w}_c^{\top} \boldsymbol{z}$. Let us denote by $D = \{(\boldsymbol{x}_i, y_i \in \mathcal{S})\}_{i=1}^N$ a mini-batch during SGD, and denote by $\nabla_{\boldsymbol{z}_i} \ell$ the gradient w.r.t. $\boldsymbol{z}_i = \boldsymbol{U}\boldsymbol{x}_i$ using the cross-entropy loss. The gradient w.r.t. the feature extractor $\boldsymbol{U}$ is thus $\nabla_{\boldsymbol{U}} \ell = \frac{1}{N} \sum_i (\nabla_{\boldsymbol{z}_i} \ell) \boldsymbol{x}_i^{\top}$. Suppose we apply SGD with a learning rate $\eta$, the updated feature extractor is $\boldsymbol{U} \leftarrow \boldsymbol{U} - \frac{\eta}{N} \sum_i (\nabla_{\boldsymbol{z}_i} \ell) \boldsymbol{x}_i^{\top}$. Building upon this formula, we can further

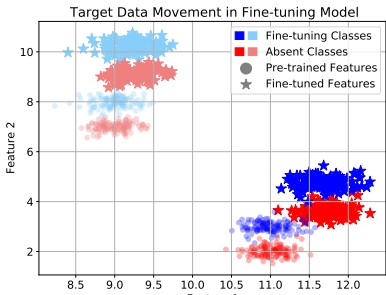

FIGURE 11. The toy example demonstrates that the features of absent class data are influenced by their similar fine-tuning class training data, resulting in absent class features moving in a similar direction as fine-tuning class features.

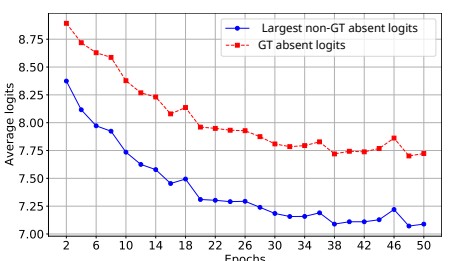

FIGURE 12. We analyze the ground-truth (GT) logits compared with the largest non-GT absent logits in ImageNet-S. While both logits decrease during the training, the consistent relative difference between them signifies a stable alignment between the features and the linear classifier.

derive how the feature $z = Ux$ of an absent class data $x$ changes after the model update

$$z \leftarrow \left( U - \frac{\eta}{N} \sum_i (\nabla_{z_i} \ell) x_i^\top \right) x = z - \frac{\eta}{N} \sum_i \nabla_{z_i} \ell (x_i^\top x). \tag{6}$$

Namely, the update of $z$ is governed by its similar training examples — those $x_i$ with high inner products $x_i^\top x$ with $x^2$ — and their corresponding feature gradients $\nabla_{z_i} \ell$. Suppose the domain shift affects similar classes similarly and the gradients w.r.t. the fine-tuning class data and features could effectively overcome the domain shift, Equation 6 offers a preliminary explanation of why fine-tuning could improve the absent class features in the downstream domain.

To further illustrate this, we design a toy example with four classes, visually depicted by different colors in Figure 11. Blue and cyan denote the fine-tuning class data; red and magenta denote the absent class data. The dimensionality of $x$ and $z$ are set to be 2; the size of $U$ is thus $2 \times 2$. We deliberately set $x$ to be non-negative to simulate the output of a ReLU operation. We create the pre-training dataset and the fine-tuning dataset by performing local translations to the data. We then pre-train a two-layer multi-layer perceptron (MLP) on the pre-training data with four classes while keeping the first layer's weight (*i.e.*, $U$) frozen as an identify matrix to ease visualization (*i.e.*, $z = x$). After pre-training, we then fine-tune the model on the downstream fine-tuning data with only two classes without freezing $U$. (Please find more details about the data creation and the model architecture in Appendix A.) As shown in Figure 11, after fine-tuning, the update of the absent class features (from ∘ to ⋆) follows the update of their closest fine-tuning class features, even though the absent class data is not involved in fine-tuning. Moreover, different classes stay quite distinguishable in the fine-tuned feature space, suggesting that fine-tuning with a subset of classes would not degrade but improve the feature quality.

**More analysis.** Due to the page limit, we leave more analyses including frozen classifiers and frozen backbones in fine-tuning, the investigation of biased logits toward fine-tuning classes, and absent class relationship analysis in Appendix C.

## 7 Conclusion

"What happens if one fine-tunes a pre-trained classifier with a subset of classes?" Prior work showed that while it improves the fine-tuning class accuracy in the downstream domain, it drastically degrades the model's ability to recognize the other classes the model had previously learned. Our systematic study, however, provides a different opinion. We found that fine-tuning does not degrade but often improves the model's ability to recognize the other classes, *if the classifiers' logits are well-calibrated.* We expect our study to serve as a valuable reference for practical fine-tuning of pre-trained models.

---

[2]Suppose $x$ and $x_i, \forall i$, are all non-negative vectors (for instance, coming from a ReLU operation on top of the prior neural network layers), their inner products will be non-negative as well.

## Acknowledgments

This research is supported by grants from the National Science Foundation (ICICLE: OAC-2112606). We are grateful for the generous support of computational resources provided by the Ohio Supercomputer Center.

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

# Appendix

We provide details omitted in the main paper.

- Appendix A: experiment and dataset details
- Appendix B: additional details for calibration
- Appendix C: additional analysis of fine-tuning
- Appendix D: detailed results of different architectures, datasets, and splits.

Our study is built upon [49]. Tu *et al*. [49] named the above the setting Holistic Transfer (HT). For ease of reference, we use the following terms interchangeably.

- fine-tuning classes & seen classes; absent classes & unseen classes;
- downstream domain & target domain; pre-training domain & source domain;
- fine-tuning & naive fine-tuning.

We emphasize that the "unseen" classes are indeed seen in pre-training but absent in fine-tuning.

## A  Experiment and Dataset Details

### A.1  Main Investigation (cf. section 3.1 in the main paper)

#### A.1.1  Dataset Details

**ImageNet-Variants**   includes ImageNet-R(endition) [15] and ImageNet-S(ketch) [54]. ImageNet-R comprises 30,000 images across 200 ImageNet classes with various renditions (*e.g*., paintings, embroidery, *etc*.) while ImageNet-S consists of 50,000 sketch-like images for 1K ImageNet classes. Each class is randomly divided into training and testing sets following an 8:2 split. 50% of the classes are randomly selected as fine-tuning classes (100 for ImageNet-R and 500 for ImageNet-S), with the remainder as absent. The downstream test set encompasses all the 200 classes for ImageNet-R and 1K classes for ImageNet-S to evaluate model performance across the full class spectrum.

**Office-Home [53]**   is a popular domain adaptation dataset, comprising 65 classes from 4 domains (Art, Clipart, Real, and Product). Following the setup in [49], Art and Real are used as pre-trained domains and each pre-trained model is then transferred to each of the three remaining downstream domains individually, resulting in six (pre-training, downstream) pairs. Within each downstream domain, each class is randomly split into training and testing sets following a 7:3 split. 30 classes are randomly selected as fine-tuning classes; the remaining 35 classes are absent classes. The downstream test set contains all the 65 classes in each downstream domain.

**VTAB [64]**   encompasses a diversity of image classification tasks. To enable zero-shot predictions with CLIP, Tu *et al*. [49] only uses the tasks that provide text names for classes: Caltech101, CIFAR100, DTD, EuroSAT, Flowers102, Pets, Resisc45, SVHN, and SUN397. Notably, the SVHN dataset was excluded from our experiments due to CLIP's difficulty in accurately predicting numerical labels as shown in [49]. Adhering to the practice in [49], we randomly sample half of the classes as tine-tuning and the remaining as absent. The downstream training set only includes images of the fine-tuning classes; the test set contains images from all classes.

Table 3 summarizes the statistics of all the datasets used in this paper.

#### A.1.2  Training Details

For the ImageNet-Variants benchmark, we use an ImageNet-1K pre-trained ResNet-50 (results in the main paper) and ViT-B/32 (results in the appendix) as pre-trained models. The pre-trained model is fine-tuned on downstream tasks for 50 epochs using the SGD optimizer with a learning rate 1e-3, momentum 0.9, weight decay 1e-4, and batch size 64. For the compared method proposed in [49], we set the hyper-parameters $\mathcal{L}_{\text{distill}} = 10$ and $\mathcal{L}_{\text{rank}} = 100$ for ImageNet-R and $\mathcal{L}_{\text{distill}} = 1$ and $\mathcal{L}_{\text{rank}} = 5$ for ImageNet-S.

TABLE 3. A statistic summary of the datasets used in this paper.

| Dataset | Pre-training domain | Downstream domain | #Classes | #Fine-tuning classes | #Downstream training | #Downstream test |
|---|---|---|---|---|---|---|
| ImageNet-Variants | ImageNet-1K | ImageNet-Rendition | 200 | 100 | 12567 | 5924 |
| | | ImageNet-Sketch | 1000 | 500 | 20444 | 10000 |
| Office-Home | Art | Clipart
Product
Real | 65 | 30 | 1,471
1,265
1,413 | 1,330
1,361
1,335 |
| | Real | Art
Clipart
Product | 65 | 30 | 857
1,493
1,459 | 750
1,330
1,361 |
| VTAB | CLIP | Caltech101
CIFAR100
DTD
EuroSAT
Flowers102
Pets
Resisc45
SUN397 | 102
100
47
10
102
37
45
397 | 51
50
23
5
51
18
22
198 | 1,371
22,513
920
8,424
510
1,445
9,159
37,542 | 6,084
10,000
1,880
5,400
6,149
3,669
6,300
21,750 |

For the VTAB benchmark, we use the ViT-B/32 CLIP models as the backbone. Specifically, we extract the class name embedding to construct the FC layer and disregard the CLIP text encoder afterward. The pre-trained model is fine-tuned on downstream tasks for 20 epochs using the SGD optimizer with a learning rate 1e-5, momentum 0.9, weight decay 0.0, and batch size 64. For the compared method proposed in [49], we follow their paper and set the hyper-parameters $\mathcal{L}_{\text{distill}} = 1$ and $\mathcal{L}_{\text{rank}} = 5$.

For the Office-Home dataset, we follow the training recipe of [49]. Firstly, the pre-trained classifier is obtained by fine-tuning an ImageNet-1K pre-trained ResNet-50 with the source domain data for 20 epochs using the SGD optimizer with a learning rate 1e-3, momentum 0.9, weight decay 5e-4, and batch size 64. After that, we fine-tuned the resulting model on each target domain for 20 epochs using the SGD optimizer with a learning rate 1e-4, momentum 0.9, weight decay 5e-4, and batch size 64. For the compared method proposed in [49], we follow their paper and set the hyper-parameters $\mathcal{L}_{\text{distill}} = 10$ and $\mathcal{L}_{\text{rank}} = 100$.

**Computational resources.** We use a combination of NVIDIA RTX A6000 and NVIDIA 2080Ti GPUs. Since we worked on fine-tuning, the computation is quite manageable.

## A.2 Ablation Study (cf. section 5 in the main paper)

In section 5 of the main paper, we investigate how 1) a biased sampling of fine-tuning and absend classes and 2) a smaller size of fine-tuning classes would impact the performance of fine-tuning. Here, we provide details of how we split the data.

In the Office-Home dataset, we strategically select fine-tuning classes that are similar in the pre-trained model's feature space, aiming to create a meaningful distinction between fine-tuning and absent classes. To identify classes that are closely related, we apply the metric of total intra-group distance: a lower value indicates higher similarity among classes within a group. This ensures finding fine-tuning classes that exhibit tight clustering in the feature space. We employ a greedy strategy to grow the fine-tuning class set towards a pre-defined size. Figure 13 ad Figure 14 present t-SNE visualizations that illustrate the distribution of chosen fine-tuning and absent classes for varying fine-tuning class sizes across two distinct pre-trained domains.

For the ImageNet-S dataset, we leverage the WordNet [11] hierarchy to select coherent groups as fine-tuning classes. Specifically, we explore two different hierarchical splits. The details are shown in Table 4.

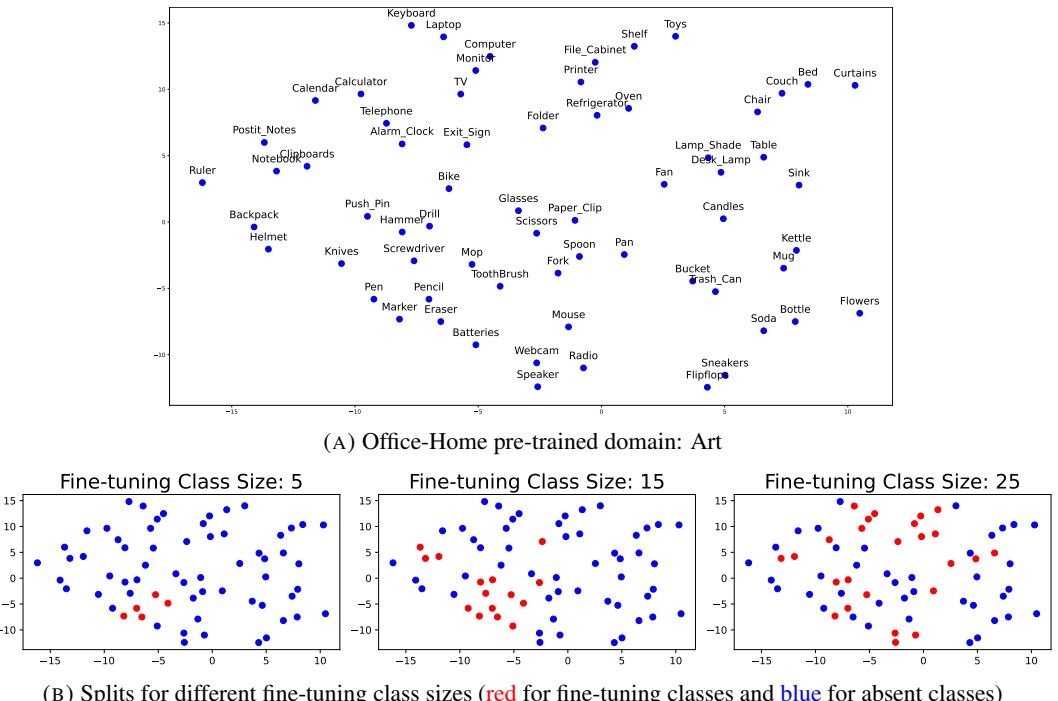

(A) Office-Home pre-trained domain: Art

(B) Splits for different fine-tuning class sizes (red for fine-tuning classes and blue for absent classes)

FIGURE 13. (A) shows the t-SNE of the class mean features for 65 classes in the Art domain with their corresponding class names. (B) shows the fine-tuning and absent classes split for different fine-tuning class sizes.

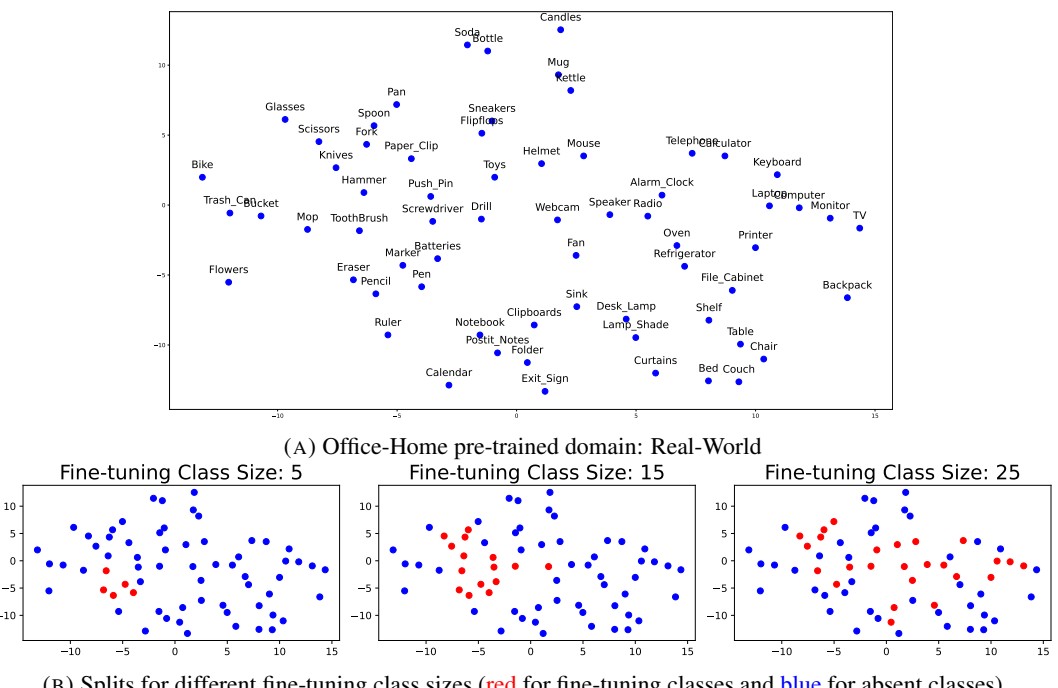

(A) Office-Home pre-trained domain: Real-World

(B) Splits for different fine-tuning class sizes (red for fine-tuning classes and blue for absent classes)

FIGURE 14. (A) shows the t-SNE of the class mean features for 65 classes in the Real-World domain with their corresponding class names. (B) shows the fine-tuning and absent classes split for different fine-tuning class sizes.

TABLE 4. Two hierarchical splits for ImageNet-S.

| Hierarchical Split One | | | |
|---|---|---|---|
| Split Group Name | Dog | Mammal | Animal |
| Split Group Size (Classes) | 118 | 218 | 398 |
| Hierarchical Split Two | | | |
| Split Group Name | Device | Instrumentality | Artifact |
| Split Group Size (Classes) | 124 | 350 | 522 |

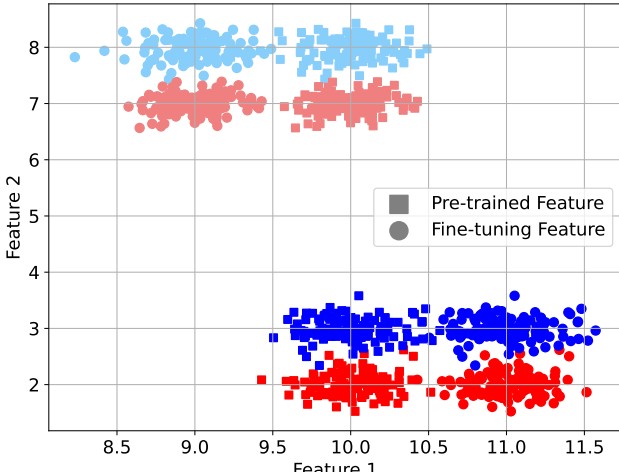

FIGURE 15. Visualization of pre-training and target domain data in the toy example, with distinct colors indicating different classes. The symbol □ represents data from the pre-training phase, while ◯ denotes those of the target domain (only blue and cyan are fine-tuning during fine-tuning).

### A.3 Toy Example (cf. section 6 in the main paper)

To elucidate the impact of similar fine-tuning training data on the feature representation of absent data, in section 6 and figure 12 of the main paper, we construct a toy example featuring four classes of 2-dimensional data, each represented by distinct colors (blue, cyan, red, magenta). Figure 15 shows the same data. In this example, pre-training data (□) are generated from Gaussian distributions with a standard deviation of 0.2 and four different means: (10, 2), (10, 3), (10, 8), and (10, 7). To simulate domain shift, the target domain data (◯) undergoes a horizontal shift, with the cyan and magenta classes moving to the right and the blue and red classes to the left by an identical distance. This setup, intentionally restricting data to non-negative values, mirrors the effect of a ReLU activation.

We employ a two-layer multi-layer perceptron (MLP) with a configuration of 2-2-4 (input dimension: 2, hidden layer dimension: 2, and output dimension: 4). The MLP is initially pre-trained on the four-class pre-training dataset, with the first layer weights set as an identity matrix to simplify visualization. Subsequent fine-tuning on target domain data incorporates only two classes (blue and cyan), without the constraint of freezing the first layer. Both the pre-training and fine-tuning phases utilize the SGD optimizer, applying a learning rate of 0.01 for 100 epochs with cross-entropy loss as the objective.

## B Additional Details for Calibration (cf. section 4 in the main paper)

We provide details for the calibration methods in section 4 of the main paper.

### B.1 Average Logit Gap (ALG)

In a well-calibrated pre-trained model, the scale of average non-Ground-Truth (non-GT) logits is expected to be similar across different classes. Figure 16 demonstrates that using the pre-trained

model, the non-GT logits exhibit comparable magnitudes for both fine-tuning and absent class groups. However, the fine-tuning class group's non-GT logits become notably higher after fine-tuning, since fine-tuning tends to assign much higher logits to see classes due to the absence of absent data. This disparity inspires an estimation of $\gamma$ based on the non-GT logit differences between fine-tuning and absent classes, as delineated in Equation 7. Concretely, for each training example, we calculate 1) the average of non-GT fine-tuning class logits and 2) the average of non-GT absent class logits. We then average the difference between them over all the training examples as an estimate of $\gamma$. The calculation is also visually summarized in Figure 17. Since this approach is based on the average non-GT logit difference between fine-tuning and absent classes, we call this method the Average Logit Gap (ALG).

$$\gamma_{\mathrm{ALG}} = \frac{1}{|D_{\mathrm{tr}}|} \sum_{(\boldsymbol{x}_i, y_i) \in D_{\mathrm{tr}}} \Big[ \frac{1}{|\mathcal{S}|} \sum_{c \in \mathcal{S} \& c \neq y_i} \boldsymbol{w}_c^{\top} f_\theta(\boldsymbol{x}_i) - \frac{1}{|\mathcal{U}|} \sum_{c \in \mathcal{U} \& c \neq y_i} \boldsymbol{w}_c^{\top} f_\theta(\boldsymbol{x}_i) \Big] \tag{7}$$

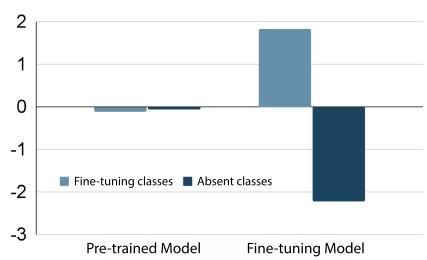

FIGURE 16. Office-Home absent test data's average non-GT logits within the fine-tuning and absent groups. In the pre-trained model, they are similar but the fine-tuning class non-GT logits become notably higher after fine-tuning.

FIGURE 17. Average Logit Gap (ALG) calibration method: the calibration factor $\gamma$ is calculated based on the difference of non-GT logits between fine-tuning and absent groups.

## B.2   Pseudo Cross-Validation (PCV)

To address the challenge of selecting $\gamma$ without access to target validation data, which ideally encompasses both fine-tuning and absent classes, we introduce a novel Pseudo Cross-Validation (PCV) method, demonstrated in Figure 18. Specifically, we partition the target training dataset $D_{\mathrm{tr}}$ into pseudo training data $D_{\mathrm{pseudo\text{-}tr}}$ and pseudo validation data $D_{\mathrm{pseudo\text{-}val}}$. $D_{\mathrm{pseudo\text{-}tr}}$ is further divided into two subsets with disjoint label spaces, simulating pseudo-fine-tuning data $D_{\mathrm{pseudo\text{-}ft}}$ and pseudo-absent data $D_{\mathrm{pseudo\text{-}absent}}$. We then naively fine-tune the pre-trained model on $D_{\mathrm{pseudo\text{-}absent}}$ and evaluate $\mathrm{Acc}_{\mathcal{S}/\mathcal{Y}}$ and $\mathrm{Acc}_{\mathcal{U}/\mathcal{Y}}$ using $D_{\mathrm{pseudo\text{-}val}}$. We select a $\gamma$ by balancing these two accuracies. To enhance the robustness of $\gamma$ estimation, we employ bootstrapping, repeating the pseudo-splitting and fine-tuning process three times with varying partitions. The selected $\gamma$ is applied to the fine-tuning model $\{\boldsymbol{\theta}_{\mathcal{T}}, \boldsymbol{W}_{\mathcal{T}}\}$, which is fine-tuned from the pre-trained model $\{\boldsymbol{\theta}_{\mathcal{O}}, \boldsymbol{W}_{\mathcal{O}}\}$ on the entire target training data $D_{\mathrm{tr}}$.

## C   Additional Analysis of Fine-tuning (cf. section 3.4 and section 6 in the main paper)

In this section, we provide more analysis including an analysis of feature improvements, an investigation of the tendency for logits to be biased toward fine-tuning classes, and the effects of freezing the classifier and backbone during fine-tuning. Additionally, we quantitatively examine the class relationship change between the pre-trained and fine-tuning models. Finally, we extend our analysis to examine the impact of fine-tuning on the iWildCam dataset [1].

### C.1   Feature Improvement by the Fine-Tuned Model

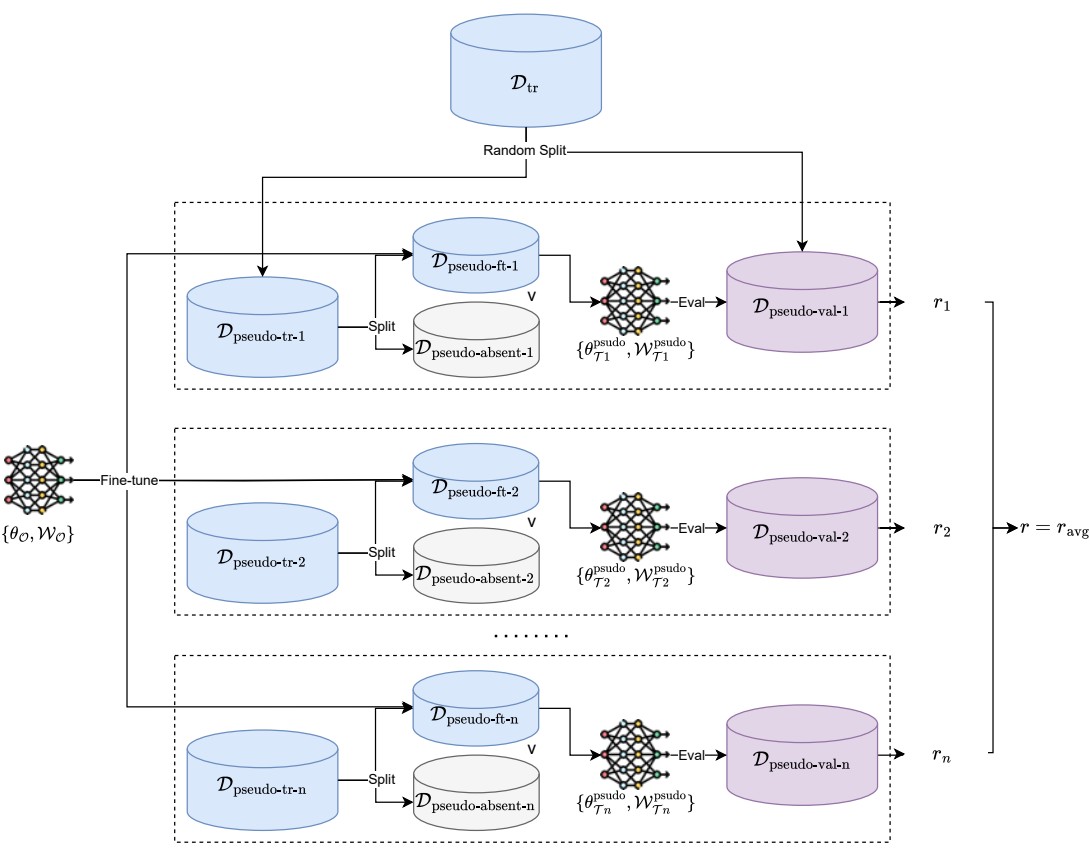

FIGURE 18. The illustration of the Pseudo Cross-Validation (PCV) method for estimating the calibration factor $\gamma$.

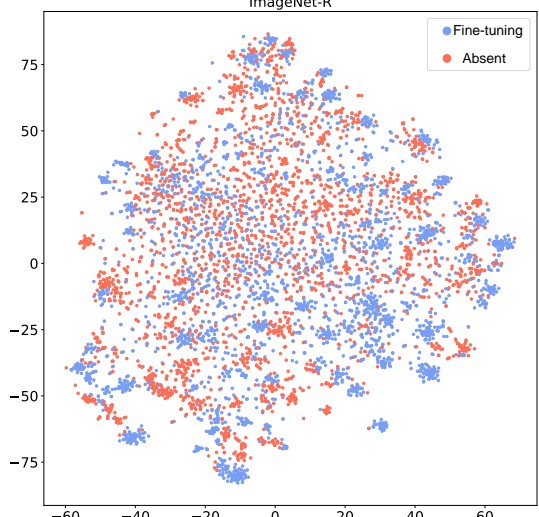

FIGURE 19. **t-SNE (ImageNet-R)**. The FT extractor $f_{\boldsymbol{\theta}_{\mathcal{T}}}$ does not create an artificial margin between fine-tuning and absent classes.

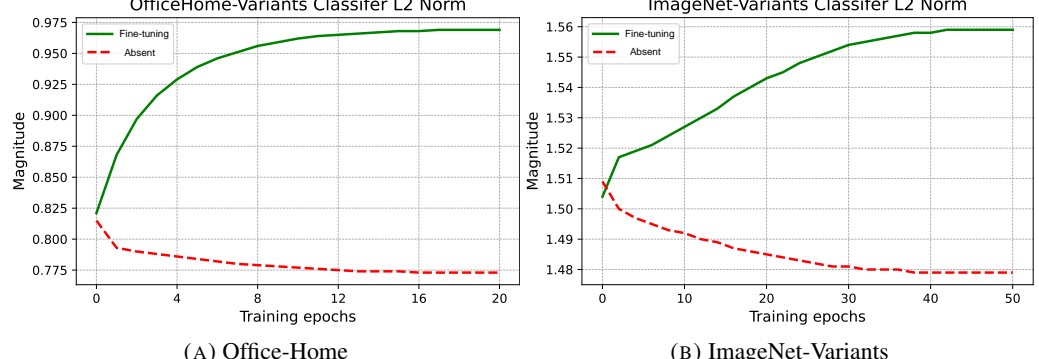

(A) Office-Home                      (B) ImageNet-Variants

FIGURE 20. L2 norm of classifier weights for fine-tuning and absent classes during fine-tuning. This illustrates a significant increase in weight magnitude for fine-tuning classes compared to absent, potentially resulting in fine-tuning's biased logits towards fine-tuning classes.

TABLE 6. The performance comparison between the neural network (NN) classifier with the FC layer and the cosine classifier.

| Benchmarks | Classifier | AUSUC | $\text{Acc}_{\mathcal{Y}/\mathcal{Y}}$ | $\text{Acc}_{\mathcal{S}/\mathcal{Y}}$ | $\text{Acc}_{\mathcal{U}/\mathcal{Y}}$ |
|---|---|---|---|---|---|
| ImageNet-Variants | NN | 0.44 | 43.35 | 81.29 | 3.53 |
| | Cosine | 0.43 | 42.65 | 79.68 | 3.77 |
| Office-Home | NN | 0.63 | 53.25 | 87.84 | 22.37 |
| | Cosine | 0.63 | 57.26 | 87.97 | 29.83 |

To further understand the improved NCM absent class accuracy NCM $\text{Acc}_{\mathcal{U}/\mathcal{Y}}(f_{\boldsymbol{\theta}_{\mathcal{T}}})$, we plot the t-SNE embedding [51] in Figure 19. We find that the FT feature extractor $f_{\boldsymbol{\theta}_{\mathcal{T}}}$ does not superficially raise $\text{Acc}_{\mathcal{U}/\mathcal{Y}}$ by creating an artificial "margin" between the fine-tuning and absent class features [3] and destroying their semantic relationship. We further report the NCM accuracy of classifying absent class data into one of the absent classes, *i.e.*, $\text{Acc}_{\mathcal{U}/\mathcal{U}}$ with $\mathcal{B} = \mathcal{U}$. As summarized in Table 5, FT indeed improves the feature quality to distinguish among absent classes.

| NCM $\text{Acc}_{\mathcal{U}/\mathcal{U}}$ | PT | FT | $\Delta$ |
|---|---|---|---|
| IN-{R, S} | 40.3% | 61.9% | 21.6% |
| VTAB | 83.7% | 87.2% | 3.5% |
| Office-Home | 80.1% | 81.5% | 1.4% |

TABLE 5. **NCM $\text{Acc}_{\mathcal{U}/\mathcal{U}}$ using the pre-trained (PT) and FT features.** FT improves the feature quality for absent classes.

### C.2   Biased Logits Towards fine-tuning Classes

section 4 of the main paper highlights how fine-tuning generates logits biased towards fine-tuning classes due to the lack of absent data during training. To delve deeper into this, we analyze the L2 norm of the classifier weights $\boldsymbol{W}_{\mathcal{T}}$ for Office-Home and ImageNet-Variants. Figure 20 reveals a pronounced increase in the weight magnitude for fine-tuning classes relative to absent classes during fine-tuning, which could potentially result in fine-tuning's biased logits towards fine-tuning classes.

To determine whether the magnitude of classification weights is solely responsible for the biased logits, we apply the cosine classifier [13], which normalizes the weights to ensure uniform magnitude across all classes. According to the results shown in Table 6, employing a cosine classifier does yield a slight improvement in absent class accuracy ($\text{Acc}_{\mathcal{U}/\mathcal{Y}}$); however, the outcome remains suboptimal. This indicates that both the magnitude of the linear classifiers and their angles (*i.e.*, cosine similarity) with the features contribute to the biased logits. These findings lay the groundwork for our proposed calibration approach, which seeks to directly adjust the logits, bypassing the limitations associated with modifying either the magnitude or the angle of the classification weights.

---

[3] Doing so would shrink the label space from $\mathcal{Y}$ to $\mathcal{U}$. We note that $\text{Acc}_{\mathcal{U}/\mathcal{U}} \geq \text{Acc}_{\mathcal{U}/\mathcal{Y}}$: the former eliminates the errors of classifying absent class samples into fine-tuning classes.

TABLE 7. Linear classifier CKA analysis results.

| Dataset | Fine-tuning Classes CKA | Absent Classes CKA |
|---|---|---|
| ImageNet-Varaints | 0.969 | 0.999 |
| Office-Home | 0.979 | 0.998 |

## C.3 Should We Freeze the Linear Classifier or Feature Backbone?

Adhering to the practice in source-free DA [29], Tu *et al.* [49] advocate for freezing the linear classifier $W_{\mathcal{O}}$ during fine-tuning (termed the frozen classifier approach) to preserve the absent class relationship and hence accuracy $Acc_{\mathcal{U}/\mathcal{Y}}$. Surprisingly, our analysis reveals that while preserving class relationships within the FC layer, the frozen classifier leads to a worse AUSUC score and NCM absent accuracy compared to fine-tuning, as evidenced in Table 8. This suggests a deterioration in the feature quality of the frozen classifier.

We hypothesize that the underlying issue arises from fine-tuning with only fine-tuning class data, which inadvertently biases the model towards classifying all samples as fine-tuning classes. While fine-tuning can adapt to this bias by adjusting the FC layer to reduce absent logits, the frozen classifier must alter its features to achieve such a goal, potentially compromising the quality of absent features.

We also investigate freezing the feature extractor backbone $\theta_{\mathcal{O}}$ and only fine-tuning the FC layer (a.k.a. linear probing). Since the feature representations remain unchanged, the NCM $Acc_{\mathcal{U}/\mathcal{Y}}$ mirrors that of the pre-trained model, indicating no improvement in feature quality, as evidenced in Table 8. As a result, it achieves a lower NCM accuracy and lower AUSUC score than fine-tuning. In sum, neither the frozen classifier nor the frozen backbone are effective strategies in HT.

## C.4 Absent Class Relationship Analysis

Inspired by the finding in [19]—the similarity among vectors in the FC layer can reflect the semantic relationships between their corresponding classes—we investigate if such class relationships are maintained during fine-tuning. To quantitatively capture the class relationships, we compute the cosine similarity between each pair of vectors in the FC layer. Specifically, the matrices $W_{\mathcal{O}}^{\mathcal{S}}(W_{\mathcal{O}}^{\mathcal{S}})^{\top}$ and $W_{\mathcal{O}}^{\mathcal{U}}(W_{\mathcal{O}}^{\mathcal{U}})^{\top}$ capture the class relationships among fine-tuning and absent classes, respectively, within the pre-trained model. Similarly, $W_{\mathcal{T}}^{\mathcal{S}}(W_{\mathcal{T}}^{\mathcal{S}})^{\top}$ and $W_{\mathcal{T}}^{\mathcal{U}}(W_{\mathcal{T}}^{\mathcal{U}})^{\top}$ reflect these relationships in fine-tuning. Here, $W$ represents the L2-normalized linear classification weights, with superscripts indicating whether the weights are for fine-tuning $\mathcal{S}$ or absent $\mathcal{U}$ classes, and subscripts differentiating between the pre-trained and the fine-tuning model.

To assess how fine-tuning affects class relationships, we employ the linear Centered Kernel Alignment (CKA) [25], to compare the class relationships before and after fine-tuning. Specifically, we compute CKA scores for fine-tuning and absent classes:

$$\text{Fine-tuning Classes CKA} = \text{CKA}(W_{\mathcal{O}}^{\mathcal{S}}(W_{\mathcal{O}}^{\mathcal{S}})^{\top}, W_{\mathcal{T}}^{\mathcal{S}}(W_{\mathcal{T}}^{\mathcal{S}})^{\top})$$
$$\text{Absent Classes CKA} = \text{CKA}(W_{\mathcal{O}}^{\mathcal{U}}(W_{\mathcal{O}}^{\mathcal{U}})^{\top}, W_{\mathcal{T}}^{\mathcal{U}}(W_{\mathcal{T}}^{\mathcal{U}})^{\top})$$

Higher CKA scores signify a more robust preservation of class relationships through the transition from a pre-trained model to fine-tuning. As demonstrated in Table 7, class relationships among absent classes are substantially more preserved than those among fine-tuning classes. This observation aligns with insights discussed in section 6 of the main paper. The distinction arises because fine-tuning with fine-tuning class data prompts the classifier to differentiate between fine-tuning classes more distinctly. In contrast, without direct training signals for absent classes, updates to absent classifiers tend to be more uniform, thus maintaining the original class relationships among absent classes throughout the fine-tuning process.

# D More Results

## D.1 ImageNet-Variants with ViT

In the main paper, our findings for ImageNet-Variants were based on experiments with a ResNet-50 model pre-trained on ImageNet-1K. To broaden our analysis, this section reports on extended

| | ImageNet-Varaints | | Office-Home | |
|---|---|---|---|---|
| Method | AUSUC | NCM Acc$_{\mathcal{U}/\mathcal{Y}}$ | AUSUC | NCM Acc$_{\mathcal{U}/\mathcal{Y}}$ |
| Pre-trained | 0.083 | 32.5 | 0.499 | 73.7 |
| Fine-Tuning | 0.439 | 50.5 | 0.632 | 75.8 |
| Frozen Classifier | 0.356 | 36.3 | 0.603 | 69.5 |
| Linear Probing | 0.218 | 32.5 | 0.595 | 73.7 |

TABLE 8. AUSUC and NCM Acc$_{\mathcal{U}/\mathcal{Y}}$ demonstrate that fine-tuning outperforms frozen classifier and linear probing.

TABLE 9. Performance comparison between ResNet-50 and ViT-B/32 in ImageNet-R with 50% fine-tuning classes (100 classes).

| Architecture | Method | AUSUC | Acc$_{\mathcal{Y}/\mathcal{Y}}$ | Acc$_{\mathcal{S}/\mathcal{Y}}$ | Acc$_{\mathcal{U}/\mathcal{Y}}$ | Acc$_{\mathcal{U}/\mathcal{U}}$ | NCM Acc$_{\mathcal{S}/\mathcal{Y}}$ | NCM Acc$_{\mathcal{U}/\mathcal{Y}}$ |
|---|---|---|---|---|---|---|---|---|
| ResNet-50 | Pre-trained | 0.09 | 23.79 | 23.42 | 24.19 | 24.61 | 37.07 | 32.68 |
| | fine-tuning | 0.46 | 43.70 | 81.97 | 1.70 | 37.92 | 73.00 | 51.63 |
| | Δ | 0.37 | 19.91 | 58.55 | -22.49 | 13.31 | 35.93 | 18.95 |
| ViT-B/32 | Pre-trained | 0.09 | 24.63 | 24.87 | 24.36 | 24.93 | 41.03 | 37.89 |
| | fine-tuning | 0.49 | 48.75 | 82.10 | 12.15 | 43.24 | 77.68 | 51.42 |
| | Δ | 0.40 | 24.12 | 57.23 | -12.21 | 18.31 | 36.65 | 13.53 |

experiments that employ the ImageNet-1K pre-trained Vision Transformer (ViT-B/32) as an alternative pre-trained model. Our objective is to ascertain whether the observations regarding fine-tuning remain consistent when applied to a larger and more advanced model architecture.

Results, as detailed in Table 9 and Table 10, indicate that both the ResNet-50 and ViT-B/32 architectures yield consistent improvements across AUSUC, Acc$_{\mathcal{U}/\mathcal{U}}$, and NCM metrics. This consistency underscores the inherent robustness of fine-tuning's benign behaviors, independent of the model architecture. Notably, the ViT-B/32 model exhibits a significantly milder decline in absent class accuracy Acc$_{\mathcal{U}/\mathcal{Y}}$ compared to ResNet-50. This finding suggests that the ViT-B/32 model is more robust in partial target data fine-tuning.

## D.2 Detailed Results for Each Dataset

In the main paper, due to space constraints, we summarized the findings by presenting average performance metrics across all datasets within each benchmark. This section expands upon that summary by providing in-depth results for each pretraining-downstream domain pair within the Office-Home dataset and for individual datasets in VTAB, detailed in Table 11 and Table 12, respectively. Across the board, fine-tuning exhibits consistent enhancements in AUSUC scores, reinforcing its efficacy in diverse settings. In ImageNet-Variants, fine-tuning consistently improves the absent class feature (an increase of NCM absent accuracy Acc$_{\mathcal{U}/\mathcal{Y}}$) as mentioned in subsection D.1. In the context of Office-Home, while the majority of domain pairs witnessed improvements in NCM Acc$_{\mathcal{U}/\mathcal{Y}}$, exceptions such as Rw-Pr and Ar-Rw experienced a decline. Some exceptions are also observed within VTAB, such as Flower102 and SUN397. These findings suggest that certain datasets may benefit from more sophisticated approaches to bolster absent class features.

## D.3 The Fine-tuning / Absent Accuracy Curve For All Datasets

Owing to the space constraints within the main document, our presentation of the Fine-tuning/ Absent Accuracy Curve was limited to ImageNet-R and the Art-Product domain pair from Office-Home. This section expands our analysis by presenting the Fine-tuning/ Absent Accuracy Curves for all datasets within each benchmark. This comprehensive display aims to offer a more detailed understanding of the performance dynamics between fine-tuning and absent classes across the diverse range of datasets evaluated in our study.

TABLE 10. Performance comparison between ResNet-50 and ViT-B/32 in ImageNet-S with 50% fine-tuning classes (500 classes).

| Architecture | Method | AUSUC | Acc$_{\mathcal{Y}/\mathcal{Y}}$ | Acc$_{\mathcal{S}/\mathcal{Y}}$ | Acc$_{\mathcal{U}/\mathcal{Y}}$ | Acc$_{\mathcal{U}/\mathcal{U}}$ | NCM Acc$_{\mathcal{S}/\mathcal{Y}}$ | NCM Acc$_{\mathcal{U}/\mathcal{Y}}$ |
|---|---|---|---|---|---|---|---|---|
| ResNet-50 | Pre-trained | 0.08 | 23.48 | 23.60 | 23.36 | 29.58 | 31.70 | 32.38 |
| | fine-tuning | 0.41 | 42.97 | 80.60 | 5.34 | 55.22 | 64.80 | 49.40 |
| | Δ | 0.33 | 19.49 | 57.0 | -18.02 | 25.64 | 33.1 | 17.02 |
| ViT-B/32 | Pre-trained | 0.06 | 21.33 | 20.78 | 21.88 | 27.32 | 30.98 | 30.46 |
| | fine-tuning | 0.39 | 46.34 | 73.38 | 19.30 | 56.92 | 64.30 | 49.24 |
| | Δ | 0.33 | 25.01 | 52.60 | -2.58 | 29.60 | 33.32 | 18.78 |

TABLE 11. Detailed results of six pretraining-downstream domain pairs in the Office-Home benchmark.

| Office-Home | Method | AUSUC | $Acc_{\mathcal{Y}/\mathcal{Y}}$ | $Acc_{\mathcal{S}/\mathcal{Y}}$ | $Acc_{\mathcal{U}/\mathcal{Y}}$ | $Acc_{\mathcal{U}/\mathcal{U}}$ | NCM $Acc_{\mathcal{S}/\mathcal{Y}}$ | NCM $Acc_{\mathcal{U}/\mathcal{Y}}$ |
|---|---|---|---|---|---|---|---|---|
| | Pre-trained | 0.30 | 47.07 | 43.65 | 50.29 | 56.14 | 57.43 | 64.04 |
| Ar → Cl | fine-tuning | 0.45 | 44.06 | 81.58 | 8.63 | 59.94 | 72.45 | 66.67 |
| | Δ | 0.16 | -3.01 | 37.93 | -41.67 | 3.80 | 15.02 | 2.63 |
| | Pre-trained | 0.53 | 67.52 | 61.32 | 71.88 | 76.5 | 80.39 | 80.25 |
| Ar → Pr | fine-tuning | 0.69 | 51.87 | 93.05 | 23.00 | 76.88 | 87.17 | 83.13 |
| | Δ | 0.17 | -15.65 | 31.73 | -48.88 | 0.38 | 6.77 | 2.88 |
| | Pre-trained | 0.62 | 72.73 | 70.67 | 74.54 | 80.87 | 77.88 | 77.78 |
| Ar → Rw | fine-tuning | 0.70 | 59.33 | 89.74 | 32.63 | 81.15 | 82.21 | 77.50 |
| | Δ | 0.09 | -13.41 | 19.07 | -41.91 | 0.28 | 4.33 | -0.28 |
| | Pre-trained | 0.51 | 65.60 | 63.06 | 68.19 | 73.32 | 69.39 | 72.24 |
| Rw → Ar | fine-tuning | 0.60 | 55.47 | 81.27 | 29.11 | 76.28 | 72.30 | 74.12 |
| | Δ | 0.09 | -10.13 | 18.21 | -39.08 | 2.96 | 2.90 | 1.89 |
| | Pre-trained | 0.33 | 51.13 | 53.22 | 49.12 | 58.70 | 64.88 | 60.77 |
| Rw → Cl | fine-tuning | 0.53 | 46.47 | 87.88 | 6.64 | 64.16 | 78.83 | 64.60 |
| | Δ | 0.20 | -4.66 | 34.66 | -42.48 | 5.46 | 13.96 | 3.83 |
| | Pre-trained | 0.70 | 78.91 | 78.60 | 79.19 | 83.38 | 84.19 | 87.01 |
| Rw → Pr | fine-tuning | 0.77 | 62.31 | 93.49 | 34.22 | 83.94 | 88.06 | 86.45 |
| | Δ | 0.07 | -16.61 | 14.88 | -44.97 | 0.56 | 3.88 | -0.56 |

TABLE 12. Detailed results of eight datasets in the VTAB benchmark.

| Dataset | Method | AUSUC | $Acc_{\mathcal{Y}/\mathcal{Y}}$ | $Acc_{\mathcal{S}/\mathcal{Y}}$ | $Acc_{\mathcal{U}/\mathcal{Y}}$ | $Acc_{\mathcal{U}/\mathcal{U}}$ | NCM $Acc_{\mathcal{S}/\mathcal{Y}}$ | NCM $Acc_{\mathcal{U}/\mathcal{Y}}$ |
|---|---|---|---|---|---|---|---|---|
| | Pre-trained | 0.70 | 78.65 | 87.97 | 70.43 | 74.85 | 86.67 | 84.23 |
| Caltech101 | fine-tuning | 0.82 | 82.27 | 97.19 | 69.10 | 85.22 | 90.18 | 88.31 |
| | Δ | 0.12 | 3.62 | 9.23 | -1.33 | 10.36 | 3.51 | 4.08 |
| | Pre-trained | 0.51 | 64.17 | 65.94 | 62.40 | 72.16 | 65.68 | 66.50 |
| CIFAR100 | fine-tuning | 0.68 | 65.90 | 89.20 | 42.60 | 78.82 | 82.00 | 74.08 |
| | Δ | 0.17 | 1.73 | 23.26 | -19.80 | 6.66 | 16.32 | 7.58 |
| | Pre-trained | 0.26 | 43.19 | 39.13 | 47.08 | 53.85 | 59.67 | 65.00 |
| DTD | fine-tuning | 0.35 | 45.32 | 75.98 | 15.94 | 50.73 | 65.22 | 65.52 |
| | Δ | 0.10 | 2.13 | 36.85 | -31.15 | -3.13 | 5.54 | 0.52 |
| | Pre-trained | 0.14 | 32.22 | 43.10 | 20.34 | 32.86 | 76.02 | 77.88 |
| EuroSAT | fine-tuning | 0.46 | 52.22 | 98.94 | 1.20 | 47.46 | 96.77 | 87.52 |
| | Δ | 0.33 | 20.00 | 55.84 | -19.14 | 14.61 | 20.75 | 9.65 |
| | Pre-trained | 0.28 | 46.71 | 44.32 | 49.08 | 55.50 | 69.12 | 72.00 |
| SUN397 | fine-tuning | 0.36 | 49.81 | 64.04 | 35.69 | 57.33 | 71.07 | 70.75 |
| | Δ | 0.08 | 3.09 | 19.72 | -13.40 | 1.83 | 1.95 | -1.24 |
| | Pre-trained | 0.37 | 53.95 | 55.24 | 52.68 | 58.83 | 81.25 | 81.67 |
| Resisc45 | fine-tuning | 0.60 | 58.48 | 93.74 | 23.66 | 66.18 | 89.14 | 81.33 |
| | Δ | 0.23 | 4.52 | 38.50 | -29.02 | 7.35 | 7.89 | -0.35 |
| | Pre-trained | 0.5 | 63.72 | 59.34 | 68.35 | 74.11 | 88.13 | 92.14 |
| Flowers102 | fine-tuning | 0.55 | 64.16 | 78.29 | 49.21 | 71.40 | 77.82 | 88.53 |
| | Δ | 0.05 | 0.44 | 18.96 | -19.14 | -2.71 | -10.32 | -3.61 |
| | Pre-trained | 0.80 | 83.97 | 79.03 | 88.60 | 94.41 | 79.93 | 77.84 |
| Pets | fine-tuning | 0.88 | 84.44 | 93.80 | 75.67 | 93.93 | 85.34 | 88.34 |
| | Δ | 0.08 | 0.46 | 14.77 | -12.93 | -0.48 | 5.41 | 10.50 |

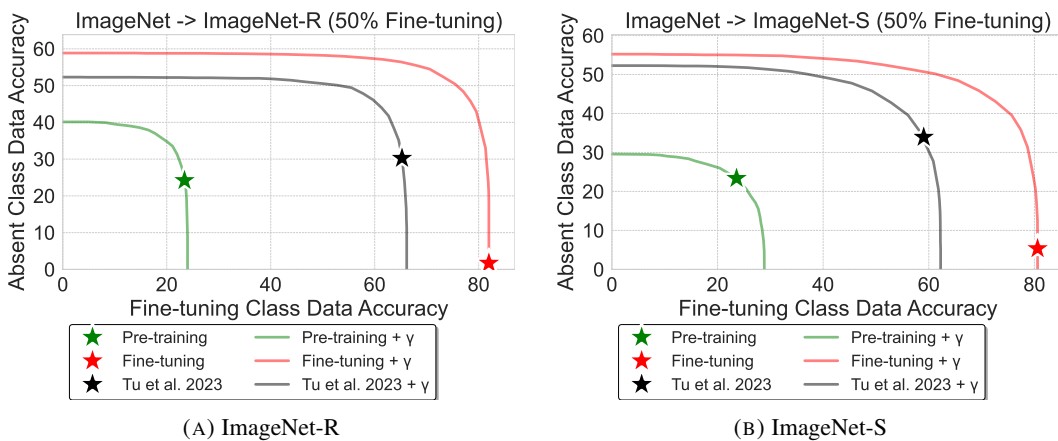

(A) ImageNet-R      (B) ImageNet-S

FIGURE 21. $Acc_{\mathcal{S}/\mathcal{Y}}$ at the x-axis and $Acc_{\mathcal{U}/\mathcal{Y}}$ at the y-axis) by varying the calibration factor $\gamma$ for ImageNet-Variants.

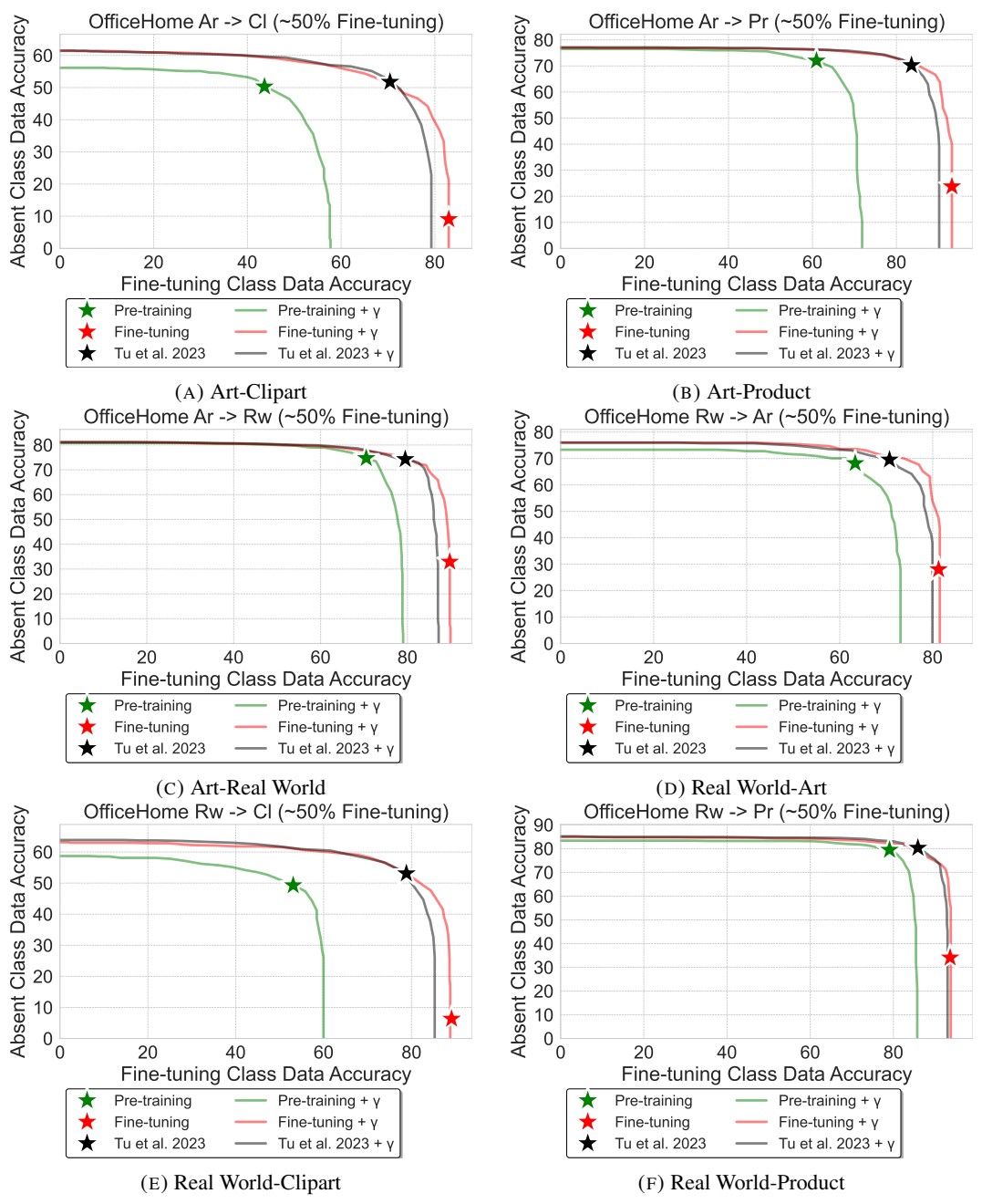

FIGURE 22. $\text{Acc}_{\mathcal{S}/\mathcal{Y}}$ at the x-axis and $\text{Acc}_{\mathcal{U}/\mathcal{Y}}$ at the y-axis) by varying the calibration factor $\gamma$ for all pretraining-downstream domain pairs in Office-Home.

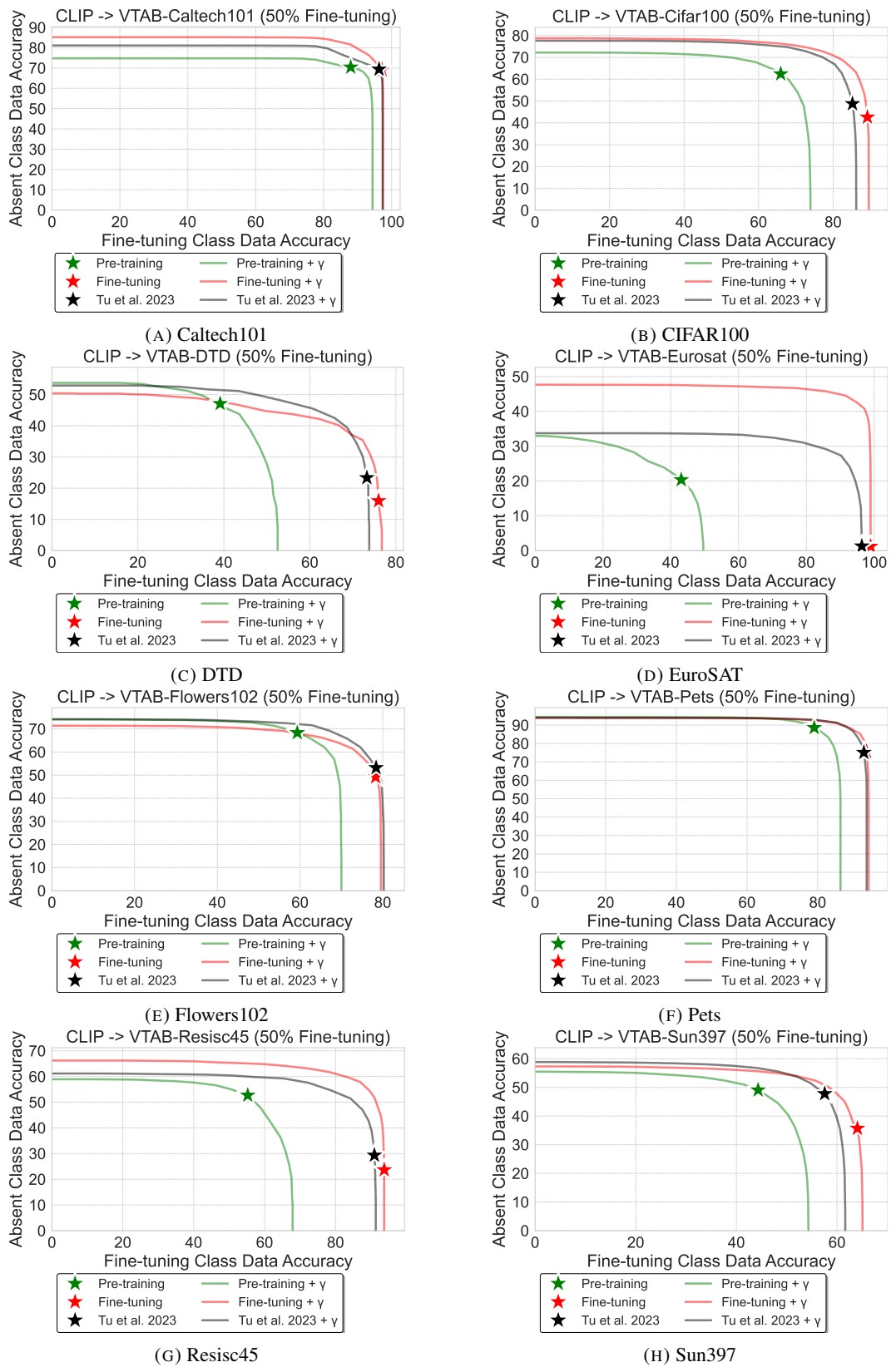

FIGURE 23. $Acc_{\mathcal{S}/\mathcal{Y}}$ at the x-axis and $Acc_{\mathcal{U}/\mathcal{Y}}$ at the y-axis) by varying the calibration factor $\gamma$ for all datasets in VTAB.

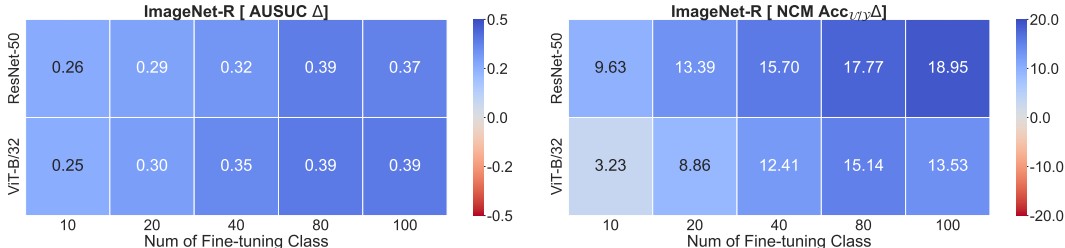

FIGURE 24. ImageNet-R: the performance gain on AUSUC and NCM $\text{Acc}_{\mathcal{U}/\mathcal{Y}}$(from pre-trained model to fine-tuning) under different data splits and fine-tuning class sizes using ImageNet-1K pre-trained models ResNet50 and ViT-B/32.

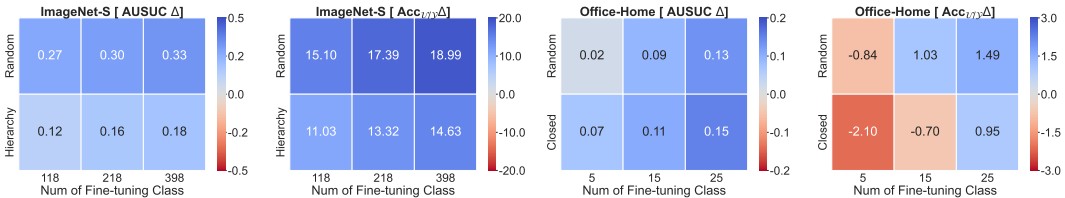

FIGURE 25. Animal hierarchical split for ImageNet-S: the performance gain on AUSUC and NCM $\text{Acc}_{\mathcal{U}/\mathcal{Y}}$(from pre-trained model to fine-tuning) under different data splits and fine-tuning class sizes using ImageNet-1K pre-trained models ResNet50 and ViT-B/32. This split contains dog (118 classes), mammal (218 classes), and animal (398 classes) as the fine-tuning classes.

## D.4 More Split Results

In the main manuscript, due to limitations on space, we restricted our presentation of data splits and fine-tuning class size-related ablation studies to ImageNet-S and Office-Home. This section extends our analysis to encompass ImageNet-R, as well as additional hierarchical splits for ImageNet-S, utilizing two distinct ImageNet-1K pre-trained models: ResNet50 and ViT-B/32.

Figure 24 delineates the outcomes for ImageNet-R where different fine-tuning class sizes are randomly chosen. Furthermore, we delve into hierarchical splits based on WordNet within ImageNet-S. Figure 25 reveals the findings from the animal hierarchical split, which encompasses dogs (118 classes), mammals (218 classes), and broader animal categories (398 classes) as fine-tuning classes. Similarly, Figure 26 presents the results from the device hierarchical split, involving devices (124 classes), instrumentality (350 classes), and artifacts (522 classes) as fine-tuning classes. Across different datasets, splits, and model backbones, the enhancements attributed to fine-tuning manifest as robust and consistent.

## D.5 Classifier Update Direction Similarity

Due to page limitations in Section 6 of the main paper, our discussion on the similarity of classifier update directions was confined to ImageNet-S. To further substantiate the universality of our findings, this section extends the analysis to the Office-Home dataset, featuring the Art-Real World domain

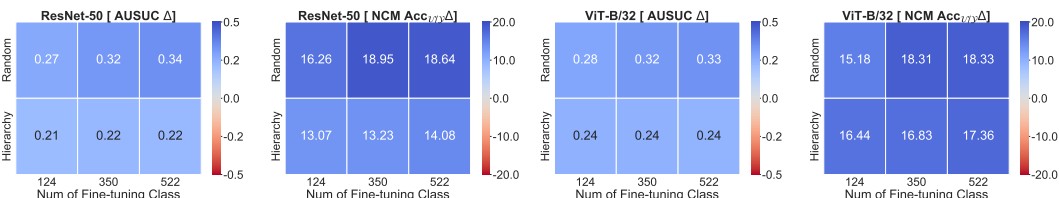

FIGURE 26. Device hierarchical split ImageNet-S: the performance gain on AUSUC and NCM $\text{Acc}_{\mathcal{U}/\mathcal{Y}}$(from pre-trained model to fine-tuning) under different data splits and fine-tuning class sizes using ImageNet-1K pre-trained models ResNet50 and ViT-B/32. This split contains device (124 classes), instrumentality (350 classes), and artifact (522 classes) as the fine-tuning classes.

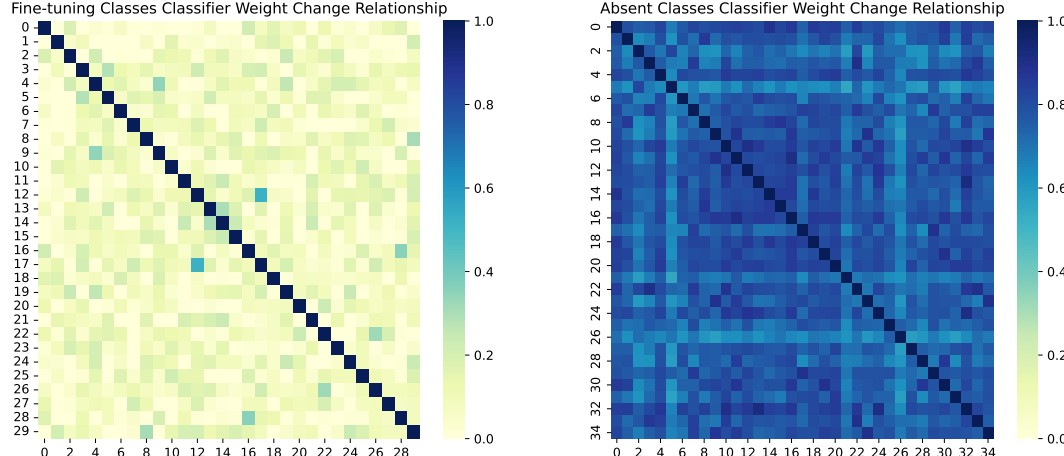

FIGURE 27. Classifier update direction similarity within fine-tuning (left) and absent (right) classes for Office-Home. The update directions are highly similar within absent classes, thus preserving the inter-class relationships among absent classes.

TABLE 13. A noticeable performance gap exists between our proposed calibration methods, ALG and PCV, and the theoretical upper limit represented by fine-tuning + $\gamma^\star$, which leverages target test data for $\gamma$ selection. This discrepancy highlights a valuable opportunity for further advancements in calibration techniques to bridge this gap.

| Metrics (%) | ImageNet-{R, S} | | | VTAB | | | Office-Home | | |
|---|---|---|---|---|---|---|---|---|---|
| | $\mathrm{Acc}_{\mathcal{Y}/\mathcal{Y}}$ | $\mathrm{Acc}_{\mathcal{S}/\mathcal{Y}}$ | $\mathrm{Acc}_{\mathcal{U}/\mathcal{Y}}$ | $\mathrm{Acc}_{\mathcal{Y}/\mathcal{Y}}$ | $\mathrm{Acc}_{\mathcal{S}/\mathcal{Y}}$ | $\mathrm{Acc}_{\mathcal{U}/\mathcal{Y}}$ | $\mathrm{Acc}_{\mathcal{Y}/\mathcal{Y}}$ | $\mathrm{Acc}_{\mathcal{S}/\mathcal{Y}}$ | $\mathrm{Acc}_{\mathcal{U}/\mathcal{Y}}$ |
| Pre-trained | 23.6 | 23.5 | 23.8 | 58.3 | 59.3 | 57.4 | 63.8 | 61.8 | 65.5 |
| fine-tuning | 43.3 | 81.3 | 3.5 | 62.8 | 86.4 | 39.1 | 53.5 | 88.3 | 22.4 |
| fine-tuning + $\gamma_{\mathrm{ALG}}$ | 55.9 | 80.3 | 30.5 | 66.8 | 85.3 | 48.2 | 65.0 | 87.7 | 44.9 |
| fine-tuning + $\gamma_{\mathrm{PCV}}$ | 57.1 | 60.1 | 54.0 | 57.4 | 47.1 | 67.8 | 72.2 | 82.3 | 63.1 |
| fine-tuning + $\gamma^\star$ | 60.8 | 73.6 | 47.6 | 69.3 | 75.6 | 62.8 | 72.7 | 79.1 | 66.9 |
| Oracle | 71.1 | 72.4 | 69.8 | 80.6 | 79.8 | 81.3 | 82.1 | 81.2 | 82.9 |

pair, as detailed in Figure 27. This pattern of update direction similarity is similarly evident across other pretraining-downstream pairings within the Office-Home dataset.

### D.6 Using Target Test Set For $\gamma$ Selection

Theoretically, utilizing the target test set for $\gamma$ selection contradicts standard practices, as it introduces data leakage that can be considered as cheating. However, for comparison purposes, we leverage the target test set to select $\gamma$, thereby denoting fine-tuning + $\gamma^\star$ as an upper bound for calibration methods. As indicated in Table 13, while our introduced calibration methods, ALG and PCV, demonstrate substantial improvements over fine-tuning, they do not fully reach the performance of fine-tuning + $\gamma^\star$. This gap underscores a direction for future research to focus on more sophisticated calibration strategies for fine-tuning.

### D.7 iWildCam

Another realistic benchmark proposed in [49] is iWildCam, which considers abundant camera trap images from various geographical locations as pre-trained domains, with images from a new camera trap location serving as the target domain. The uniqueness of this benchmark lies in the use of data collected within a limited time frame (e.g., the first month) at the new location as the target training set. Given the unlikely presence of all animal species within this initial period, the target training data inherently exhibits a class bias towards the species that are detected.

Contrary to other benchmarks, as depicted in Table 14, fine-tuning shows a decline in both AUSUC and $\mathrm{Acc}_{\mathcal{U}/\mathcal{U}}$. This performance drop is attributed to the natural data collection bias caused by time in iWildCam. Unlike conventional benchmarks that assume training and test sets come from the same distribution, iWildCam's unique structure—where training and testing sets originate from different

TABLE 14. Performance of fine-tuning on iWildCam benchmark.

| Method | AUSUC | $\text{Acc}_{\mathcal{Y}/\mathcal{Y}}$ | $\text{Acc}_{\mathcal{S}/\mathcal{Y}}$ | $\text{Acc}_{\mathcal{U}/\mathcal{Y}}$ | $\text{Acc}_{\mathcal{U}/\mathcal{U}}$ | NCM $\text{Acc}_{\mathcal{S}/\mathcal{Y}}$ | NCM $\text{Acc}_{\mathcal{U}/\mathcal{Y}}$ |
|---|---|---|---|---|---|---|---|
| Pre-trained | 0.156 | 35.79 | 36.347 | 25.888 | 33.881 | 43.965 | 31.986 |
| fine-tuning | 0.13 | 36.5 | 53.52 | 0.595 | 27.256 | 44.075 | 31.778 |
| Δ | -0.026 | 0.71 | 17.173 | -25.293 | -6.625 | 0.11 | -0.208 |

time periods—introduces a natural discrepancy in data distribution due to seasonal changes or animal migration.

To investigate this further, we selected four target locations from the benchmark and identified the pre-trained and test data corresponding to the same target training classes for each location. By extracting features using a pre-trained model, we aimed to determine whether significant changes occurred within the same classes across pre-trained data, target training data, and target testing data. The findings, illustrated in Figure 28, reveal notable differences in the data distributions among the same classes across the pre-trained domain, target training, and target testing sets. This discrepancy introduces further complexity in HT. Addressing this challenge, future research may focus on enhancing fine-tuning with strategies capable of bridging the data distribution gap.

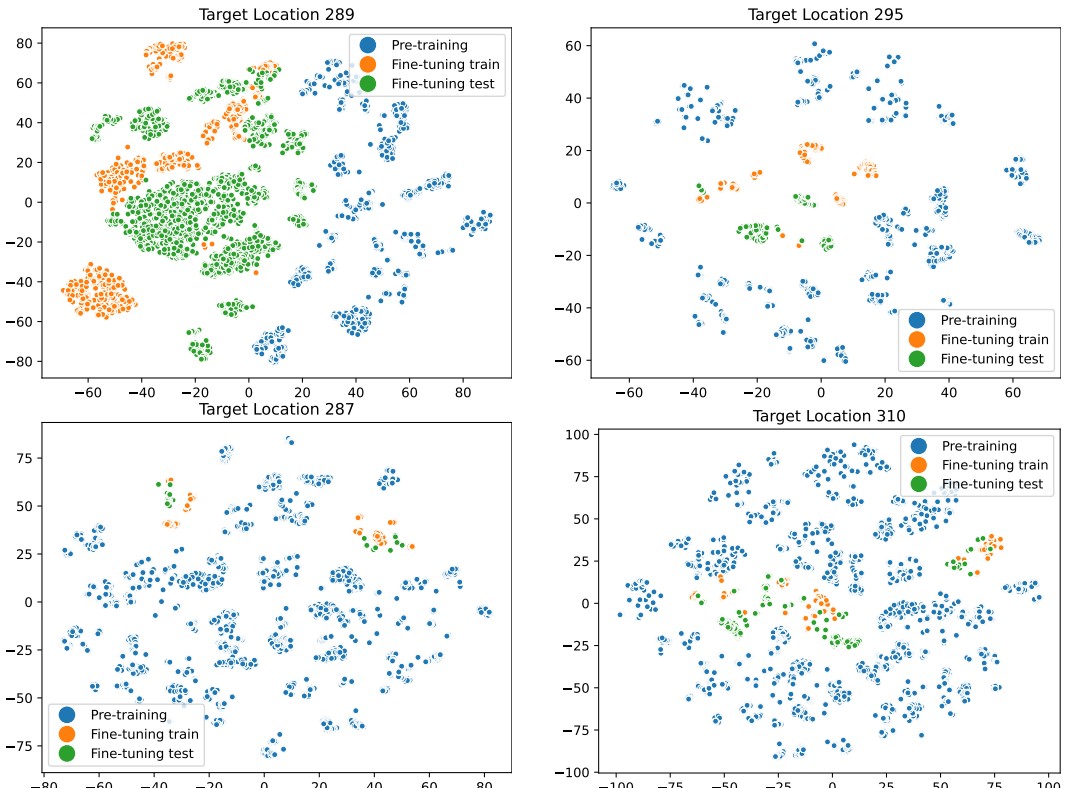

FIGURE 28. iWildCam t-SNE visualization of the same group of classes (target training classes) in pre-trained domain data, target training data, and target testing data, demonstrating significant distribution differences. This variation presents an additional challenge in HT. The finding is consistent across 4 different target locations.

