# OpenReview forum: "Fine-Tuning is Fine, if Calibrated"
_NeurIPS.cc/2024/Conference — NeurIPS 2024 poster_

### Official Review · Reviewer_YzGv · 2024-07-01

**Soundness:** 3
**Presentation:** 3
**Contribution:** 3
**Rating:** 5
**Confidence:** 4

**Summary:**

The paper proposes a simple post-training calibration technique for classifying missing classes after fine-tuning. For example, assuming the pre-trained model can classify 1000 classes and is fine-tuned on a subset of these classes from a different image domain, the proposed method improves the classification accuracy of the *absent classes* on this new domain. Specifically, the proposed method adds a calibration hyper-parameter to artificially boost the probability of predicting the absent classes.  To motivate the method, the paper investigates the quality of feature learning using the Nearest Class Mean classifier to isolate the cause of bad performance on the absent classes. While simple, the method shows good performance gain on multiple datasets.

**Strengths:**

* The proposed method is easy to implement and provides good performance gains.
* The paper uses the NCM classifier to investigate the feature extractor's quality and isolate the linear classifier's influence after fine-tuning.  This methodology provides a clear motivation for the proposed method.
*  The finding that fine-tuning does not completely destroy the features of absent classes is interesting. This insight can motivate further study, especially for improving the robustness of fine-tuning.
* The paper provides a detailed ablation study showing the proposed method's strengths and limitations.
  * The extent of performance gain hinges on the fine-tuning procedure.
  * The distribution of absent classes also affects performance.

**Weaknesses:**

* **Statements are not precise**. In the abstract and introduction, the paper claims that a fine-tuned model does not forget the relationship among absent classes. However, this claim is not precise. As the paper points out, the extent of forgetting and degradation depends on the fine-tuning procedure. For example, an Adam optimizer with a larger learning rate can degrade the features of the absent class. This is consistent with the existing literature on the robustness of fine-tuning.  It's possible that the proposed technique only works well under moderate changes to the pre-trained model.

* **The fine-tuning setting is limited**. The proposed method only works for a particular fine-tuning configuration under constrained assumptions. Specifically, the method assumes a classification task, and the pre-trained model can classify all fine-tuning classes, including absent ones. Therefore, the paper's claims on forgetting and feature learning are limited by its scope.

**Questions:**

* While the paper investigated several factors that can affect calibration effectiveness, it is unclear how we should decide when the calibration is useful. Could the authors consider dependency on the model's **intrinsic** properties? The external factors all may lead to a common intrinsic property. For example, maybe the fine-tuned model's deviation (in the weight space) can be an indicator.  For example, RMSprop and Adam are known to converge faster and potentially lead to a larger deviation from the pre-trained model?

**Limitations:**

The paper does not have a potential negative societal impact.

---

> ### Author Rebuttal · Authors · 2024-08-07
>
> We appreciate your detailed review and positive assessment of the strengths of our paper. We address your concerns as follows.
>
> Weakness:
> **W1: Statements are not precise. … The extent of forgetting and degradation depends on the fine-tuning procedure. ….**
>
> We apologize if our statement in the abstract and introduction were not precise or clear. In our humble opinion, learning rates are hyper-parameters of optimizers, and our statement was made assuming the hyper-parameters are properly selected. As mentioned in Lines 31-38, our study was motivated by the findings in [48]. We note that in [48], SGD is the main optimizer and *the learning rate has been carefully chosen.* However, the absent class accuracy still suffers a drastic drop, and [48] viewed it as an instance of forgetting. Our statement in the abstract, *“To our surprise, we find that the fine-tuned model neither forgets the relationship among the other classes nor degrades the features to recognize these classes,”* was made against the claim by [48]. We found that the accuracy drop is mainly due to the biased logits, not the forgetting of features and class relationships for the absent classes.
>
> We hope the paragraph above clarifies your concern.
>
> In our humble opinion, when the learning rate is not well chosen or the optimizer is not applied properly, any observations about the machine learning model may be misleading or doubtful.
> For full disclosure, we indeed (re-)learned such a lesson when we extended our study beyond the SGD optimizer used in [48] (Lines 72 - 76 and Lines 316 - 325). When we first applied the Adam optimizer, we followed the practice of setting a small learning rate (e.g., 1e-3 and 1e-4) to ensure that the training could converge (on the fine-tuning classes). Under such a setting, we saw a notable gain in the fine-tuning class accuracy but a poor NCMu/y and ACCu/u (thus a poor AUSUC), contracting our findings (Lines 42 - 64) when using the SGD optimizer. At that time, *we almost drew the conclusion that our finding is optimizer-specific.* However, by further reducing the learning rate, we once again saw a decent NCMu/y and ACCu/u and a similar gain in AUSUC as using SGD (see Figure 9). This motivates us to conduct a comprehensive analysis, exploring a wide range of learning rates across six optimizers.
>
> **Q1: It's possible that the proposed technique only works well under moderate changes to the pre-trained model.** We appreciate your question. As our goal is to preserve (and even improve) the discriminative ability on the absent classes from the pre-training model to the fine-tuning models, it is reasonable that the change to the pre-trained model cannot go arbitrarily large. With that being said, the increase in AUSUC in Figure 9 (e.g., from 0.5 to 0.63) suggests that the fine-tuned model has undergone a sufficient change to improve its performance in the downstream domain.
>
> **Q2. The fine-tuning setting is limited.**
> We acknowledge that our scope is limited to classification tasks. However, we respectfully think it is a limitation but not necessarily a weakness. In our humble opinion, many advances in machine learning start from the investigation of classification problems (e.g., AlexNet and self-supervised learning) and gradually extend to other tasks. While we work on a particular fine-tuning configuration with a subset of classes, we respectfully think it is a practical setting (with access to large pre-trained models like CLIP but limited fine-tuning data) that deserves deeper exploration. With this in mind, we conducted a systematic, extensive analysis on a focused topic. While we agree that exploring other tasks could broaden the applicability of our findings, the focused scope of our current study allows us to thoroughly investigate and validate our claims.
>
> **Q1: Dependency on the model's intrinsic properties.**
> Thank you for the insightful question. As mentioned in Lines 85 - 98 and Lines 253 - 257, the calibration method works when the drop in the absent class accuracy primarily comes from the biased logits, not the degradation of the feature extractor or the relationship among absent classes.
>
> We follow your suggestion to explore the model's intrinsic properties. Since the parameter space of the model is of extremely high dimensionality, it is hard to use the L2 distance between the pre-trained and fine-tuned models to quantify the model deviation. We thus explore a different measurement.
>
>
> We calculated the linear Centered Kernel Alignment (CKA) in Appendix C.5 (Lines 733-737) and found that the CKA of the absent classes’ linear classifiers (between the pre-trained and fine-tuned models) correlates well with performance after calibration. As shown in Figure R.1 of our rebuttal PDF, the Unseen CKA across different optimizers follows a similar pattern to their Area Under the Seen-Unseen Curve (AUSUC) in Figure 9. Specifically, when the Unseen CKA falls below a certain threshold (e.g., 0.98), there is a corresponding drastic drop in AUSUC. This suggests that the CKA can be a useful indicator of calibration effectiveness. We will complete and include this study in the camera-ready version.

---

> > ### Author Response · Authors · 2024-08-12
> > **Kindly request your response**
> >
> > Dear Reviewer YzGv,
> >
> > We appreciate your valuable comments on our paper. We have prepared a rebuttal (together with a general response to all reviewers) and tried our best to address most if not all of your concerns. We notice that the author-reviewer discussion period is coming to an end, and we are willing to answer any unresolved or further questions that you may have regarding our rebuttal if time is allowed.
> >
> > If our rebuttal has addressed your concerns, we would appreciate it if you would be willing to consider raising your original rating. Thank you for your consideration.
> >
> > Best,
> > Authors

---

> > > ### Comment · Reviewer_YzGv · 2024-08-13
> > >
> > > Thank the authors for the extended discussion. I will keep my score because the scope of the paper is limited.

---

> > > > ### Author Response · Authors · 2024-08-14
> > > > **Re: Official Comment by Reviewer YzGv**
> > > >
> > > > Thank you for the prompt response. We are glad that you keep the positive score. We will incorporate the rebuttal into our final version.

---

### Official Review · Reviewer_5T9R · 2024-07-08

**Soundness:** 2
**Presentation:** 1
**Contribution:** 3
**Rating:** 5
**Confidence:** 4

**Summary:**

The authors argue that fine-tuning doesn’t forget the features for classes not participating in it, but rather downscales their logits as a result of which the model ends up being overconfident for the fine-tuning classes. Counter-intuitively, the authors claim that fine-tuning also enhances the discriminative ability of the model for the classes not participating in fine-tuning. For this, the authors analyze that accuracy of the NCM classifier on the features of the model and show that it increases even for classes absent during fine-tuning. Then the authors show that the order of the absent classes is still in place amongst the set of absent classes, but the model becomes overconfident on the classes used for fine-tuning, thereby leading to drop in accuracy of absent classes. To fix this, the authors analyze simple post processing calibration methods and demonstrate recovery in the performance of absent classes, with some drop in performance of fine-tuning classes. The analysis is done on Imagenet-R, VTAB and Office-Home datasets.

**Strengths:**

1) The gains observed on calibrating the model are impressive, and it is interesting to see that merely by calibrating the model, the model’s performance on absent classes can improve to this extent.
2) The motivation behind the post hoc calibration method is interesting and it is unexpected that just the confidence of the absent would lower down while preserving their relative order on performing fine-tuning.

**Weaknesses:**

1) An increase in NCM classifier’s accuracy need not necessarily mean that the model has become better in discriminating features. It only means that the features become closer to the corresponding class mean in l2 distance metric space. I think this argument is not concrete enough and requires more evidence.
2) It is not clear, why a drop in accuracy is seen in the fine-tuning classes on using PVC as the post calibration method. Even in case of ALG, where there isn’t a significant drop, the accuracy of classes absent during fine-tuning is still significantly lower (e.g. on office home it drops by over 20%) than the pre-trained model. This suggests that authors claim on feature enhancement of absent classes might not be true.
3) The authors propose to use training data / validation data to find the right threshold. This would mean access to the model as well as data. In such a scenario someone could rather easily fine-tune the model on absent classes as well, which would not require a lot of compute. Therefore, on a practical standpoint, it is not completely obvious how the proposed method benefits more than mere fine-tuning. It would be great if authors could compare the budget required for fine-tuning on absent classes vs post-hoc calibration to address this point to achieve similar performance. Although I agree that the observation itself is interesting, but analyzing the efficiency aspect could help the authors in making the claim on using post calibration methods stronger.

4) I think that the analysis shown in Fig-9 needs more rigor. Further, I think the authors need not discuss this in the main paper, as it doesn't adds much. The learning rate used for different optimizers are same, but the learning rates for adam, adagrad and adagrad need to be scaled down to make a fair comparison with SGD.
5) The analysis on why absent class features improve on fine-tuning is not rigorous and would encourage authors to not discuss this in the main paper. Most of the claims in this analysis seem low hanging and not sound enough. This hinders the readily of the paper currently.
6) Similarly, figure-10 doesn't adds much to the storyline authors have presented in this work and it requires more rigor. I would suggest them to restructure the paper a bit and remove this analysis from the main paper.

Minor comments:
In figure-5, the y-axis should be between 0-1 since it represents a probability.

I would be happy to increase my scores if my concerns are sufficiently addressed.

**Questions:**

It would be great if the authors can present their results on domainnet dataset [1].
I request the authors to kindly address the questions in the weaknesses section.

[1] https://paperswithcode.com/dataset/domainnet

**Limitations:**

Yes, the authors have addressed the limitations.

---

> ### Author Rebuttal · Authors · 2024-08-07
>
> We thank the reviewer for the valuable comments. We are pleased that the reviewer recognize several key strengths of our paper. We respond to your concerns (weaknesses and questions) as follows.
>
> **W1. NCM classifier’s accuracy and discriminating features.**
>
> Thank you for the comment. Our paper focuses on classification problems. Thus, we consider improvement in *classification accuracies* as one critical metric to assess the *features’ discriminative abilities*. We note that increasing the NCM accuracy requires the features of each class not only to be close to the corresponding class but also to be far away from other classes, meaning that the class means need to be separated apart as well. These properties align with linear discriminative analysis and, in our humble opinion, are what we expect discriminative features to possess.
>
> Besides NCM, we note that in section 4.3 ( specifically Figure 4), the fine-tuned model’s accuracy among absent classes also increases. We view it as another evidence that the features’ discriminative ability improves.
>
> With that being said, we are aware that NCM is not the only way to assess feature quality, and we will be happy to include further analyses.
>
>
> **W2: Accuracy by PCV and ALG.**
>
> We respectfully think there might be a misunderstanding. Our main finding in the paper is that with proper calibration of the logits, the fine-tuned model (red $\star$ in Figure 2) can regain and even increase its accuracy in the absent classes (the y-axis value along the red curve can surpass the green $\star$), suggesting that its features on absent classes do not degrade but often improve. (Please also see our remark in Lines 94-98.)
>
> We note that the post-processing calibration factor $\gamma$ is applied after the model has been fine-tuned. In other words, the accuracy drop in either the fine-tuning or absent classes (as the reviewer found) may result from sub-optimal selections of $\gamma$ and have nothing to do with feature qualities. Indeed, when $\gamma$ is not properly set, either the fine-tuning class accuracy or the absent class accuracy can drop drastically, to zero, corresponding to the two endpoints of the red curves in Figure 2 and Figure 7.
> PCV and ALG are two approaches to selecting $\gamma$, but we certainly do not claim they are optimal. In Table 1, we can see that “Fine-tuning + $\gamma^\star$” could achieve a much better balance of ACCu/y and ACCs/y than “Fine-tuning + $\gamma$PCV” and  “Fine-tuning + $\gamma$ALG.” We note that “Fine-tuning + $\gamma^\star$” outperforms the pre-trained model in ACCu/y and ACCs/y, indicating the fine-tuned models have enhanced accuracy in both categories, if calibrated properly. We leave a better, more robust approach to select $\gamma$ in our future work.
>
> **W3: Training data/validation data to find the right threshold.** We respectfully think there might be a misunderstanding. We apologize if we did not describe the setting clearly and we will certainly improve it.
>
> As mentioned in Section 5 of the main paper, “Pseudo cross-validation (PCV) partitions D_{tr} into **pseudo**-fine-tuning and **pseudo**-absent classes and finds $\gamma$ that can balance the pseudo-fine-tuning and pseudo-absent class accuracy.” We note that D_{tr} is not the data used in pre-training (which contains absent class data), but the fine-tuning data, as defined in section 3 (Lines 150 - 155). The **pseudo**-fine-tuning and **pseudo**-absent classes are both from the fine-tuning classes; no absent class data are exposed to PCV. (More details can be found in section B.2 of the appendix.)
>
> **W4: Figure 9 and learning rates.**
> We will improve Figure 9 to make it easier to read. We respectfully think that our study in section 6 (Lines 316 - 325) is valuable and essential, as we want to know whether our findings in sections 4 and 5 are general or optimizer-specific. Regarding the learning rate, we note that the values along the x-axis in Figure 9 are not the exact learning rates, but the multiplying factors to the **default learning rate (LR)** of each optimizer, respectively (please see Lines 319 - 320). We obtain the default LR from the `torch.optim` and `pytorch-optimizer` packages. We presented their values in Table R.3 in the rebuttal pdf, and we will include it in the camera-ready version. We apologize that we did not make this part clear in the main paper. After all, Figure 9 is not meant to argue that SGD is better than other optimizers, but to show that our findings in sections 4 and 5 hold for different optimizers when their learning rates are properly set.
>
> **W5 & 6: Analysis (Lines 326 - 360) and Figure 10.**
> Thank you for your feedback. We apologize if these parts were not rigorous, and we will consider moving them to the Appendix. We acknowledge that the analysis is not a formal theory, and we will certainly clarify it. We note that we have conducted several other analyses to understand our findings (c.f. Line 361 - 363). Due to the space limit, we keep them in the Appendix (section C and section D.5). We will polish and condense some of them to replace the current Lines 326 - 360 in the main paper.
>
> **W7: Figure 5.** We will adjust the y-axis. Thank you for pointing it out.
>
> **Q1: DomainNet**
> As per your suggestion, we conducted additional experiments on DomainNet. We followed setting 1 in section 4.1 to pre-trained a Resnet-50 model on the ‘Real’ domain with 345 classes and then fine-tuned it on the randomly selected 170 classes in other five domains(ClipArt, Sketch, Infograph, Painting, and Quickdraw). We include the results in the rebuttal PDF (Table R.1 and Table R.2).
>
> We can see that average AUSUC (c.f. Lines 280 - 289 in the main paper) and ACCu/y both increase after fine-tuning, even though absent classes have not been involved in the fine-tuning process, which is consistent with our findings from other datasets. We will include the complete results and discussions in the camera-ready version.

---

> > ### Author Response · Authors · 2024-08-12
> > **Kindly request your response**
> >
> > Dear Reviewer 5T9R,
> >
> > We appreciate your valuable comments on our paper. We have prepared a rebuttal (together with a general response to all reviewers) and tried our best to address most if not all of your concerns. We notice that the author-reviewer discussion period is coming to an end, and we are willing to answer any unresolved or further questions that you may have regarding our rebuttal if time is allowed.
> >
> > If our rebuttal has addressed your concerns, we would appreciate it if you would be willing to consider raising your original rating. Thank you for your consideration.
> >
> > Best,
> > Authors

---

> > ### Comment · Reviewer_5T9R · 2024-08-13
> >
> > Thanks to the authors for their rebuttal. The rebuttal mostly addresses my concerns and therefore, I will increase my score.

---

> > > ### Author Response · Authors · 2024-08-14
> > > **Re: Official Comment by Reviewer 5T9R**
> > >
> > > We are glad that our rebuttal has addressed most of your concerns and you are willing to increase the score. We will incorporate the rebuttal into our final version. Thanks.

---

### Official Review · Reviewer_9Ei3 · 2024-07-09

**Soundness:** 4
**Presentation:** 4
**Contribution:** 4
**Rating:** 8
**Confidence:** 4

**Summary:**

It is commonly believed that fine-tuning zero-shot models on seen classes will lead to a decrease in performance on unseen classes.
In this paper, the authors systematically examine the issue that find that (1)  the fine-tuned feature extractor is not damaged: NCM improves the absent class accuracy without catastrophic forgettin (2) main factor that damages the FT model’s ability to correctly classify absent class examples is the biased logit values towards fine-tuning classes. (3) a simple post-processing calibration of logits, ie, offsets the seen classes could bring back the zero-shot performance of absent classes. Extensive experiments and analyses validate the claims of the authors.

**Strengths:**

The paper is well-organized and the presentation is clear. The analyses are comprehensive and convincing. This study provides insights that corrected my previous viewpoint that fine-tuning causes the forgetting of knowledge of absent classes. This paper is undoubtedly a valuable work and worth accepting.

**Weaknesses:**

This is a solid paper, and I did not find any major weaknesses.

I suggest that the authors include more experiments on CLIP models, such as the base-to-new setting in CoCoOp, reporting AccY/Y, AccS/Y, and AccU/Y on 11 datasets.

**Questions:**

Please refer to the weaknesses.

**Limitations:**

No negative societal impact has been identified.

---

> ### Author Rebuttal · Authors · 2024-08-07
>
> Thank you for the positive feedback. This project has been a challenging journey, yet rewarding. We respond to your valuable comment below.
>
> **W: “I suggest that the authors include more experiments on CLIP models, …”**
>
> Thank you for the suggestion. We will certainly include more experiments and discussions on CLIP models. Compared to conventional classification models that *learn for each class a linear classifier* on top of the feature extractor, CLIP models *learn a shared “text encoder”* that can generate linear classifiers (i.e., text embeddings) given class names or descriptions. We surmise that such a shared encoder would facilitate fine-tuning with a subset of classes. Concretely, instead of directly updating the linear classifiers as in conventional classification models, fine-tuning a CLIP model would update the “text encoder” that generates the linear classifiers. Given only the fine-tuning data from a subset of classes, the CLIP model has the potential to learn their common properties (e.g., domain specifics) and transfer them to the absent classes through the shared encoder. As a result, the discrepant logit scales might be reduced. The prompting capability of CLIP models further enables new approaches like CoCoOp to adapt the model without adjusting its parameters. We will perform a detailed and systematic study on fine-tuning with CLIP models, considering both full fine-tuning and CoCoOp and reporting ACCy/y, ACCs/y, and ACCu/y. (We note that our fine-tuning classes correspond to CoCoOp’s base classes; our absent classes align with its new classes.) We will also investigate whether calibration is compatible with CLIP models to further boost the fine-tuning performance.

---

> > ### Comment · Reviewer_9Ei3 · 2024-08-08
> >
> > Thank you for your reply. I am interested in reproducing the results presented in your paper, but I noticed that your code was not provided and the code of the baseline [1] is not publicly available. Could you let me know if there are any plans to release the code, and if so, when it might be available?
> >
> > Best,
> >
> > [1] holistic transfer: towards non-disruptive fine-tuning with partial target data

---

> > > ### Author Response · Authors · 2024-08-12
> > > **Re: Official Comment by Reviewer 9Ei3**
> > >
> > > Dear Reviewer 9Ei3,
> > >
> > > We appreciate your interest in our study and in reproducing the results presented in our paper. We understand the importance of reproducibility in research and are committed to supporting it. We plan to release our code along with the camera-ready version of the paper. We appreciate your understanding and patience.
> > >
> > > Best,
> > > Authors

---

### Official Review · Reviewer_amJX · 2024-07-13

**Soundness:** 3
**Presentation:** 3
**Contribution:** 3
**Rating:** 6
**Confidence:** 4

**Summary:**

The paper unveils the improved features of absent classes when a pre-trained model is fine-tuned on a subset of all classes. The paper presents an empirical study on three datasets to demonstrate this finding and proposes a calibration method to post-process the logits after fine-tuning to improve the classification result in absent classes. The reason why the absent classes are improved is analyzed and the effectiveness of the calibration method is supported by experimental results.

**Strengths:**

1. The finding on the improved performance of absent classes is interesting.


2. The proposed calibration method is simple and easy to use.


3. The presentation of the paper is clear.

4. The empirical results in the appendix are extensive.

**Weaknesses:**

1. The reason why the absent classes' features are improved is demonstrated to be that the fine-tuned classes have similar features as absent classes. That suggests that the improvement does not always hold when the fine-tuned classes have features that are not helpful or even harmful (e.g., spurious correlation) to absent classes. This should be discussed further in the submission.

2. In some figures it looks like Tu et. al. achieves the best trade-off, while the proposed scaling method is only presented with a line. It is probably better to show the performance of the two $\gamma$ selection methods in the figure.

**Questions:**

NA

---

> ### Author Rebuttal · Authors · 2024-08-07
>
> Thank you for your positive assessment and valuable feedback on our work.
>
> **Q1: … the improvement does not always hold when the fine-tuned classes have features that are not helpful or even harmful ...**
>
> Thank you for your insightful question, and we will include more discussions in our camera-ready version.
>
> In our study, while not explicitly mentioned, we suppose the pre-trained model has learned a decent discriminative ability and faithful similarity among classes (in the pre-training domain), and we aim to preserve and even improve these properties (in the downstream domain) after fine-tuning with a subset of classes.
>
> Our analysis in Section 6 (Lines 326 - 360) explains why the absent class features could improve after fine-tuning. *We note that we certainly did not claim that the improvement would always happen under any conditions.* For instance, in Lines 336 - 339, we pointed out one necessary condition. Essentially, if the domain shift affects similar classes *differently* — for example, huge domain shifts break the similarity learned in the pre-trained model — then the improvement would unlikely hold. We will extend this discussion to incorporate non-helpful or harmful features. For example, if the domain shift introduces spurious correlations so that dissimilar classes (e.g., some fine-tuning and absent classes in the pre-training domains) have similar features in the downstream domain, then the features after fine-tuning might become misleading.
>
> After all, just like domain adaptation techniques typically degrade when there is a huge discrepancy between the source and target domains, we think it is reasonable that our findings or proposed approach may not work in some (extreme) conditions. We will surely extend the discussion so that readers or future users can get a better understanding of when the improvement may or may not hold.
>
>
> **Q2: … it looks like Tu et. al. achieves the best trade-off, while the proposed scaling method is only presented with a line. …**
>
>
> Thank you for the suggestion and we will include the two $\gamma$ selection methods in the figures where appropriate.
>
>
> Meanwhile, we want to reiterate the *difference* and *compatibility* between Tu’s method and our calibration method. We note that Tu’s method is a fine-tuning approach that updates the pre-trained model, and in our paper, we compare it to no fine-tuning and conventional full fine-tuning. The three $\star$ in Figure 2 and Figure 7 of the main paper correspond to them: black for Tu’s method; green for no fine-tuning; and red for conventional full fine-tuning. (All of them are without calibration yet.)
>
>
> In contrast, the calibration method with $\gamma$ is to adjust the strengths of the logits of the fine-tuning and absent classes, and it can be applied as post-processing to all the aforementioned models: it creates the three Seen-Unseen Accuracy Curves (AUSUCs) in Figure 7.
>
>
> In our paper, our main finding is that with calibration, the conventionally fully fine-tuned model (red) can regain its Absent class accuracy and can potentially achieve a better balance of Fine-tuning and Absent class accuracies than Tu’s method (black), as evidenced by that the red curves surpass the black curves in many of the figures (c.f., Figure 7, Figure U, Figure V, and Figure W).
>
>
> We appreciate your detailed observations in several figures in Figure V, where the black $\star$ (Tu’s method, with more complex training than full fine-tuning) seems to surpass the red curves or achieve the best trade-off that one can obtain on the red curves. We will certainly add the selected $\gamma$ on the red curves. Besides that, we note that the calibration method can be used to improve Tu’s method as well, as shown in Figure U and Figure W. With an appropriate $\gamma$, we could obtain a better trade-off along the black curves than the black $\star$.

---

> > ### Author Response · Authors · 2024-08-12
> > **Kindly request your response**
> >
> > Dear Reviewer amJX,
> >
> > We appreciate your valuable comments on our paper. We have prepared a rebuttal (together with a general response to all reviewers) and tried our best to address most if not all of your concerns. We notice that the author-reviewer discussion period is coming to an end, and we are willing to answer any unresolved or further questions that you may have regarding our rebuttal if time is allowed.
> >
> > If our rebuttal has addressed your concerns, we would appreciate it if you would be willing to consider raising your original rating. Thank you for your consideration.
> >
> > Best,
> > Authors

---

### Author Rebuttal · Authors · 2024-08-07

We thank the reviewers for their valuable comments. We are glad that the reviewers found our findings and motivations “interesting” (amJX, 5T9R, YzGv), our study providing “insights” correcting prior belief (9Ei3), our solution “simple and easy to use” (amJX, YzGv) with “impressive” gains (5T9R), our empirical results “extensive” (amJX, 9Ei3, YzGv), our analyses “comprehensive and convincing” (9Ei3, YzGv), and our presentation “clear” and “well-organized” (amJX, 9Ei3). Reviewer 9Ei3 further said, “This paper is undoubtedly a valuable work and worth accepting.” Reviewer YzGv also said, “This insight can motivate further study, especially for improving the robustness of fine-tuning.”

We have tried our best to address most if not all of the concerns, and we will modify our camera-ready version accordingly to incorporate all the feedback.

---

### Decision · Program_Chairs · 2024-09-25

**Decision:**

Accept (poster)

**Comment:**

The paper presents a nice observation that "forgetting" of absent classes during finetuning might occur due to poor calibration rather than actually destroying information about classes. Based on this, the paper provides a method to finetune that does not result in this forgetting of absent classes. The experiments show good gains, and all the reviewers are generally positive. While this is overall a valuable contribution, the main weaknesses are around on the scope of the results: unclear how it applies to CLIP-style models with text encoders that are more popular currently, and only applies to cases where we know apriori the set of absent classes we care about (even if we don't need any explicit data access). This restricts the scalability/generality of this approach, but the findings and method are nevertheless interesting and hence I recommend acceptance.